# UBE2J2 sensitizes the ERAD ubiquitination cascade to changes in membrane lipid saturation

Aikaterini Vrentzou [1], Florian Leidner [2], Claudia C. Schmidt[1,3], Helmut Grubmüller[2] & Alexander Stein [1] ✉

Protein–lipid crosstalk is fundamental to homeostasis in the endoplasmic reticulum (ER). The ER-associated degradation (ERAD) pathway, a branch of the ubiquitin–proteasome system, maintains ER membrane properties by degrading lipid metabolic enzymes. However, the ERAD components that sense membrane properties and their mechanisms remain poorly defined. Using reconstituted systems with purified ERAD factors, we show that membrane composition modulates the ubiquitination cascade at multiple levels. The membrane-anchored E2 UBE2J2 acts as a sensor for lipid packing: in loosely packed membranes, UBE2J2 becomes inactive due to membrane association that impedes ubiquitin loading, while tighter packing promotes its active conformation and interaction with E1. UBE2J2 activity directs ubiquitin transfer by the E3 ligases RNF145, MARCHF6, and RNF139, targeting both themselves and the substrate squalene monooxygenase. Additionally, RNF145 senses cholesterol, altering its oligomerization and activity. These findings reveal that ERAD integrates multiple lipid signals, with UBE2J2 relaying and extending the effect of lipid signals through its cooperation with multiple E3 ligases.

Biological membranes are complex assemblies of proteins and lipids. Each membranous subcellular compartment exhibits a unique membrane composition necessary to sustain specific biochemical activities[1–3]. Different lipid species and their relative abundance influence the dynamics of intermolecular interactions and define collective membrane properties, such as lipid packing and membrane fluidity[4]. The endoplasmic reticulum (ER) comprises a continuous membrane network characterized by a low content of phospholipids with saturated fatty acyl chains (SFAs) and low cholesterol and sphingolipid concentrations[5,6]. The resulting loose packing and low rigidity of the ER membrane are optimal for fundamental ER functions such as membrane protein biogenesis[7–10], lipid biosynthesis[11,12], and trafficking[13,14]. Intricate regulatory networks maintain the specific properties of the ER membrane. Such adaptive responses require the

integration of signals from multiple sensors that monitor various membrane characteristics. Sensor output eventually controls lipid metabolism via changes in gene expression, for example through the membrane property sensors Ire1[15,16] and Mga2[17,18], and/or adjustments in activity and stability of mediators of lipid metabolism by post-translational mechanisms.

An important post-translational mechanism regulating protein activity is ER-associated protein degradation (ERAD). ERAD executes key homeostatic functions by targeting ER-resident lipid biosynthetic enzymes for degradation, thus adjusting metabolic activity to cellular demand (for recent reviews see[19–21]). Specific lipids and collective membrane properties affect ERAD in several ways. First, they influence recruitment of substrates to membrane-embedded ubiquitin ligases (E3s). For example, the key enzymes in the mevalonate pathway for

[1]Research Group Membrane Protein Biochemistry, Max Planck Institute for Multidisciplinary Sciences, Göttingen, Germany. [2]Department of Theoretical and Computational Biophysics, Max Planck Institute for Multidisciplinary Sciences, Göttingen, Germany. [3]Present address: ETH Zürich, Zürich, Switzerland.
✉e-mail: alexander.stein@mpinat.mpg.de

cholesterol biosynthesis, squalene monooxygenase (SQLE) and 3-hydroxy-3-methylglutaryl-coenzyme A reductase (HMGCR), interact with E3s in a lipid-dependent manner. Specifically, SQLE interacts with and is ubiquitinated by the ERAD complex MARCHF6/UBE2J2 in membranes enriched in SFAs and sterols or deprived of its substrate squalene[22–27]. HMGCR recruitment to and ubiquitination by several E3s, including RNF145 and RNF139, is enhanced by sterol metabolites and mediated by INSIG adaptor proteins (for review see[28]). Second, recent work has shown that elevation of specific ceramide species impairs ERAD by inhibiting substrate extraction[29]. Third, the membrane composition modulates the enzymatic activity of ERAD components. Cellular uptake of certain lipids, for instance, increases auto-ubiquitination activity and reduces the stability of several E3 ligases. High cholesterol levels result in increased auto-ubiquitination and concomitant destabilization of the E3 RNF145[30,31] and its homolog RNF139[32,33], but have the opposite effect on MARCHF6[34]. Besides cholesterol, administration of saturated fatty acids (SFAs) causes destabilization of RNF145[31].

How changes in membrane composition are sensed by the ubiquitination machinery and eventually translated into changes in ubiquitination activity is unknown. Most models suggest that membrane-embedded ERAD E3 ligases directly sense membrane composition. For instance, for RNF139 and RNF145 the presence of a sterol-sensing domain (SSD) similar to that of HMGCR or SCAP has been proposed[30,33,35–37]. However, E3s are not the only membrane-associated components in the ubiquitination cascade. Most ubiquitin-conjugating enzymes (E2s) involved in ERAD also associate with the ER membrane, either directly or indirectly. UBE2J2 (Ubc6 in yeast) and UBE2J1 are anchored in the membrane via C-terminal transmembrane segments (TMS)[38–41], whereas UBE2G2 (Ubc7 in yeast) is membrane- associated via the integral membrane protein AUP1 (similar to Cue1 in yeast)[42–44]. Whether E2 activity is sensitive to specific lipids or modulated by global membrane properties has not been investigated. E2 enzymes interact with RING E3 ligases, which mediate the direct transfer of ubiquitin from the E2 enzyme onto the protein substrate. While E3 ligases determine substrate specificity, E2s determine chemoselectivity and linkage specificity of the ubiquitin attachment reaction. For example, UBE2G2 mediates formation of Lys48-linked ubiquitin chains and modifies substrate lysines[45], whereas UBE2J2—and probably UBE2J1—additionally modify serine and threonine residues, thus expanding the substrate repertoire[24,39,46]. They usually serve as priming E2s that mediate attachment of the first ubiquitin onto the substrate[40,47,48].

In cellular systems, transcriptional and post-translational homeostatic responses are often difficult to disentangle experimentally, as they are both regulated by lipids[30,31,49]. In addition, in sequential reactions such as the ubiquitination cascade, membrane properties can affect any membrane-associated enzyme. In this work, we reconstitute purified human ERAD components into membranes of defined lipid composition to investigate how specific lipids and global membrane properties regulate ubiquitination. We dissect the ubiquitination cascade into individual reactions and find that lipid packing directly modulates the activity of the E2 enzyme UBE2J2, whereas cholesterol specifically influences the E3 ligase RNF145. We show that regulation of E2 activity has broad downstream effects on multiple E3 ligases, impacting both E3 auto-ubiquitination and substrate ubiquitination. These findings reveal a previously unrecognized lipid-dependent regulation of a ubiquitin-conjugating enzyme and suggest that multi-layered regulation of the ubiquitin cascade contributes to maintaining membrane homeostasis.

## Results

### UBE2J2 is inactive in ER-like membranes
In order to dissect the effect of different membrane compositions on the ERAD ubiquitination cascade, we first investigated loading of

UBE2J2 and UBE2J1 with ubiquitin by the ubiquitin-activating enzyme (E1). To this end, we reconstituted purified and fluorescently labeled full-length E2s into phospholipid vesicles (liposomes) with a low fraction of saturated acyl chains on phospholipids and low cholesterol concentrations that mimicked the ER membrane[5,6] (Supplementary Fig. 1a–c). To follow ubiquitin-loading, we incubated these liposomes with ubiquitin (Ub), E1, and ATP, and analyzed samples by non-reducing SDS-PAGE, which maintains the E2-Ub thioester linkage (Fig. 1a). Ubiquitin loading of UBE2J2 in ER-like membranes was slow and occurred only over an extended period of incubation (Fig. 1b, c and Supplementary Fig. 1d, e). In contrast, in detergent solution, UBE2J2 was almost completely loaded within 1 min (Fig. 1b, c), and was additionally auto-ubiquitinated, mostly on cysteine residues other than the catalytic Cys94 (Supplementary Fig. 1f, g), similar to the auto-ubiquitination observed for the yeast homolog Ubc6[48,50]. Suppression of UBE2J2 loading by membrane reconstitution was not due to incorrect UBE2J2 orientation, as the majority of UBE2J2 was correctly oriented in ER-like membranes (Supplementary Fig. 1h, i), and thus principally accessible to the ubiquitination machinery. Also, low UBE2J2 activity was not due to reduced E1 activity in the presence of liposomes, as high lipid concentrations in the reaction did not affect thioester bond formation between ubiquitin and E1 (Supplementary Fig. 1j). Furthermore, E1 concentration was not limiting at the concentrations used in our experiments (Supplementary Fig. 1k). UBE2J2 inactivation in the membrane was reversible, as re-solubilization of proteoliposomes restored UBE2J2 loading (Supplementary Fig. 1l, m).

Next, we tested ubiquitin loading of the other membrane-anchored, structurally similar E2, UBE2J1. Unlike UBE2J2, UBE2J1 was efficiently loaded with ubiquitin in both detergent solution and ER-like membranes, with the correctly oriented fraction fully loaded within 30 s (Fig. 1d, e; Supplementary Fig. 1n). These data indicate that membrane-dependent inhibition of E2 activity is specific to UBE2J2.

### Membrane properties tune UBE2J2 activity
As UBE2J2 was largely inactive in an ER-like membrane, we hypothesized that changes in membrane composition could activate it and thus increase ERAD activity. The ER membrane is defined by a high content of unsaturated acyl chains and resulting membrane packing defects. Lipidomic data on ER-derived vesicles from *Saccharomyces cerevisiae* and mammalian cells, showed that SFAs represent about 25–35% of total acyl chains, with SFAs rarely occupying the sn-2 position, and 50–60% of the phospholipids containing unsaturated acyl chains in both the sn-1 and sn-2 positions[5,6]. To assess phospholipid saturation effects on UBE2J2, we compared its ubiquitin loading after reconstitution in liposomes with the same phosphatidylcholine (PC): phosphatidylethanolamine (PE) molar ratio (80:20), but drastically differing acyl chain composition (Supplementary Fig. 2a, b). Similar to ER-like membranes that contained 33% SFAs, UBE2J2 activity was low in membranes with 10% SFAs (Fig. 2a, b). In stark contrast, membranes made solely of palmitoyl-oleyl phospholipids (POPL, 50% SFAs) showed a marked increase in the fraction of UBE2J2 immediately loaded by E1, even at low E1 concentration (Supplementary Fig. 2c). Raising SFA content to 60% by replacing some POPC with di-palmitoyl PC (DPPC) did not further enhance activity. This activation of UBE2J2 was not due to the decrease in acyl chain length accompanied by replacing oleate (18:1) with palmitate (16:0), but rather due to acyl chain saturation (Supplementary Fig. 2d, e).

To resolve the rapid kinetics of UBE2J2 loading in membranes with high SFA content, we used a quenched-flow apparatus. In membranes with 60% SFAs, a large fraction of UBE2J2 was ubiquitin-loaded within tens of milliseconds, while a second pool was only getting loaded on the minute timescale. By contrast, the fast-loading pool was almost absent in membranes with 10% SFA content (Fig. 2c). These results fit a

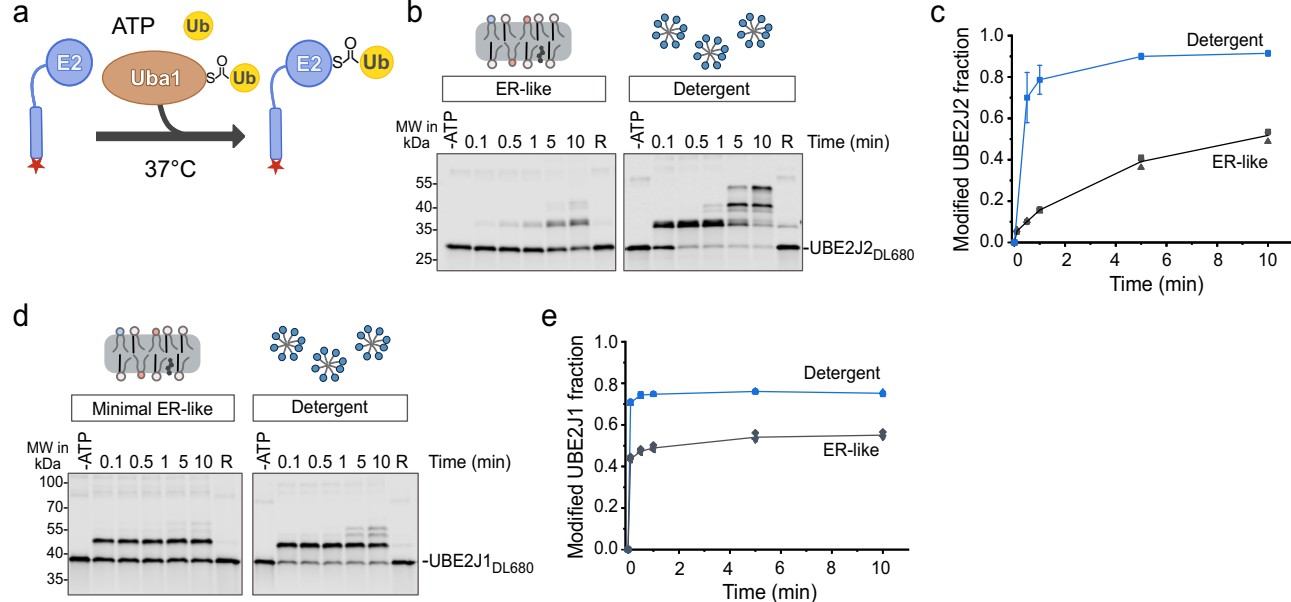

**Fig. 1 | E2 enzymatic activity is low in ER-like membranes. a** Schematic of ubiquitin loading assay. Fluorescently labeled E2s were incubated at 37 °C with ubiquitin-loaded E1 (Uba1) in the presence of ATP and free ubiquitin. UBE2J1 and UBE2J2 both possess an N-terminal UBC domain connected to the C-terminal transmembrane segment by a disordered region. The red star marks the position of the fluorescent label. **b** UBE2J2$_{DL680}$, reconstituted into ER-like liposomes (protein: lipid ratio 1:32,000) or dissolved in detergent, was subjected to the reaction described in (**a**). Samples were collected at indicated time points, separated by non-reducing SDS-PAGE, and analyzed by fluorescence scanning. Where indicated, ATP was omitted (-ATP) or the 10 min sample was reduced with 2% (v/v)

2-mercaptoethanol (R). **c** Quantification of ubiquitin-loaded UBE2J2$_{DL680}$ from reactions as in (**b**). Lines connect mean values. For reactions with more than three replicates, error bars indicate the mean ± standard deviation ($n$ = 7, detergent; $n$ = 3, ER-like liposomes). Each replicate for liposome conditions represents an independent protein reconstitution, and for detergent conditions an independently set up reaction. **d** Ubiquitin loading assay as in (**b**), but with UBE2J1$_{DL680}$. **e** Quantification of ubiquitin-loaded UBE2J1$_{DL680}$ from reactions as in (**d**) ($n$ = 3, both conditions). R reduced. ER-like liposomes contained 60 mol% POPC, 20 mol% DOPE, 10 mol% DOPS, 10 mol% cholesterol. Complete lipid compositions are given in Table 1. Source data are provided as a Source Data file.

simple kinetic model, in which UBE2J2 exists in two states, an active and an inactive one; membrane composition affects the interconversion between these states, while the reactivity of the active conformation toward E1 is unaffected by the membrane (Fig. 2d). Discharge of the loaded state was not accounted for in this model, as it occurs on a minute timescale and is unaffected by membrane composition (Supplementary Fig. 2f, g).

To further define the sensitivity of UBE2J2 to membrane composition, we incrementally replaced POPL with di-oleyl phospholipids (DOPL). Compared to pure POPL membranes, UBE2J2 activity decreased in membranes with 40% SFAs (20% DOPL) and was further reduced at 35% SFAs (30% DOPL) (Fig. 2e, f). These findings indicate that UBE2J2 is sensitive to small deviations from the reported physiological content of unsaturated lipids.

We also analyzed the related E2 UBE2J1 to assess whether lipid saturation effects are a general feature of membrane-anchored E2s. In assays comparing membranes with 10% vs. 60% SFAs, the fraction of UBE2J1 immediately available for ubiquitin loading was only modestly lower in the high-SFA condition (Supplementary Fig. 2h, i). However, antibody-accessibility assays showed that UBE2J1 orientation varied substantially with membrane composition (Supplementary Fig. 2j), at least partially accounting for the small differences observed in loading. While we did not investigate UBE2J1 kinetics or lipid sensitivity further, these results indicate that the pronounced lipid saturation effect is specific to UBE2J2 and not a general property shared with UBE2J1.

UBE2J2 may sense changes in membrane composition either through direct interactions with saturated acyl chains or indirectly via saturation-induced changes in global membrane properties. To assess the effect of saturated acyl chains on lipid packing of liposomes, we used the fluorescent probe C-Laurdan, a sensor for lipid

packing at the membrane plane close to the lipid headgroup[51]. Dehydration of the membrane due to tighter lipid packing results in a blue-shifted emission of C-Laurdan, expressed as higher generalized polarization. In PC/PE membranes, gradually increasing the content of SFAs resulted in higher C-Laurdan polarization, i.e., tighter lipid packing (Fig. 2g). Thus, UBE2J2 activation correlated with increased lipid packing. To investigate whether UBE2J2 is a membrane property sensor, we induced changes in lipid packing by manipulating the PE and cholesterol content. Compared to PC/PE liposomes, PC-only membranes showed reduced generalized polarization, confirming that PE increases packing[18,52]. Accordingly, UBE2J2 inactivation in unsaturated membranes was even stronger in the absence of PE (Fig. 2h, i). However, PE was not required for activation of UBE2J2 in membranes with high SFA content, arguing against a specific effect of its headgroup.

Adding 10 mol% cholesterol to PC/PE liposomes increased generalized polarization of C-Laurdan (Supplementary Fig. 2k) and slightly enhanced UBE2J2 activity in membranes with 35% SFAs, but had no effect at 60% SFAs (Fig. 2j, k). The magnitude of the cholesterol-induced increase in C-Laurdan polarization was disproportionate to its effect on UBE2J2 activity. We attribute this lack of correlation, at least in part, to the limited ability of C-Laurdan to accurately probe membrane hydration in the presence of cholesterol[53].

Dynamic light scattering showed that variations in membrane composition did not affect (proteo-) liposome size, indicating that differences in membrane curvature do not account for changes in E2 activity (Supplementary Fig. 2l, m). Furthermore, protein presence had no significant impact on C-Laurdan polarization (Supplementary Fig. 2n). Together, these findings reveal that UBE2J2 is sensitive to changes in its membrane environment and suggest that it is a sensor for lipid packing.

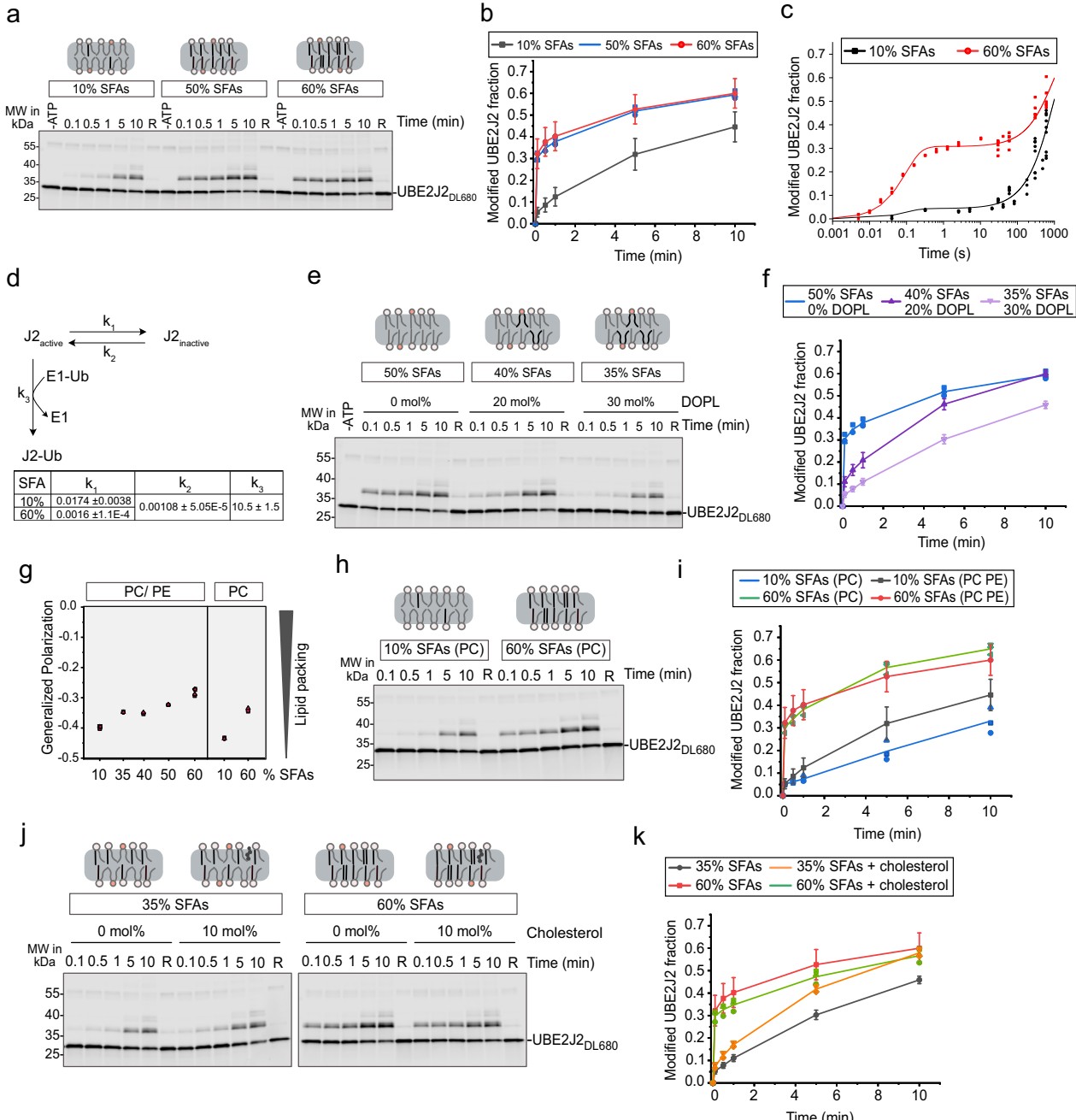

**Fig. 2 | Membrane composition dictates the efficiency of UBE2J2 loading.**
**a** Ubiquitin loading assay, with UBE2J2$_{DL680}$ reconstituted into liposomes containing 80 mol% PC and 20 mol% PE with indicated saturated fatty acids (SFA) content. **b** Quantification of assays as in (**a**); $n = 7$ (10% SFAs), $n = 9$ (60% SFAs), $n = 3$ (50% SFAs). Solid lines connect mean values; for $n > 3$, error bars represent mean ± s.d., standard deviation. **c** Kinetic analysis of ubiquitin loading of UBE2J2 by quenched-flow. Data from (**b**) is included in the plot and used for the fitting procedure. Solid lines represent the fitting result. **d** Schematic of the kinetic model used for the fitting procedure. The table below shows the result of the fitting procedure with units s[-1]. **e** As in (**a**), but with liposomes varying in di-unsaturated lipid content (P/L = 1:32,000). **f** Quantification of reactions as in (**e**). $n = 3$ (50% SFAs), $n = 4$ (40% and 35% SFAs). **g** Lipid packing of protein- and sterol-free liposomes with various SFA contents measured by C-Laurdan fluorescence. Emission spectra were

recorded at 37 °C ($n \geq 3$); generalized polarization was calculated as described[18]. **h** Ubiquitin loading of UBE2J2$_{DL680}$, reconstituted in PC-only liposomes with indicated SFA content (P/L = 1:32,000). **i** Quantification of reactions as in (**h**). Data for PC-only liposomes are shown in blue and green ($n = 3$); for comparison, data from (**b**) obtained with PC/PE liposomes are shown black and red. **j** Ubiquitin loading of UBE2J2$_{DL680}$, reconstituted in PC/PE liposomes with indicated SFA content with or without 10 mol% cholesterol (P/L = 1:32,000). **k** Quantification of reactions as in (**j**). Mean values are connected by solid lines; $n = 2$ (35% SFAs + cholesterol), $n = 4$ (35% SFAs, no cholesterol, reproduced from (**f**)), $n = 9$ (60% SFAs, no cholesterol, reproduced from (**c**)), $n = 3$ (60% SFAs + 10% cholesterol). In (**b**), (**f**), (**i**) and (**k**), each replicate represents an independent protein reconstitution into liposomes. For $n > 3$, error bars represent mean ± s.d. Detailed liposome compositions are given in Table 1. Source data are provided as a Source Data file.

## Lipid packing affects UBE2J2 conformation

Next, we sought to characterize the molecular mechanism of UBE2J2 inactivation in membranes with packing defects. We hypothesized that membrane properties affect UBE2J2 conformation. We performed

limited proteolysis assays on proteoliposomes with either 10% or 60% SFAs using trypsin, and detected N-terminally truncated UBE2J2 fragments that fluoresce at their C-terminus (Fig. 3a). In both membrane types, trypsin generated a ~10 kDa fragment corresponding to the

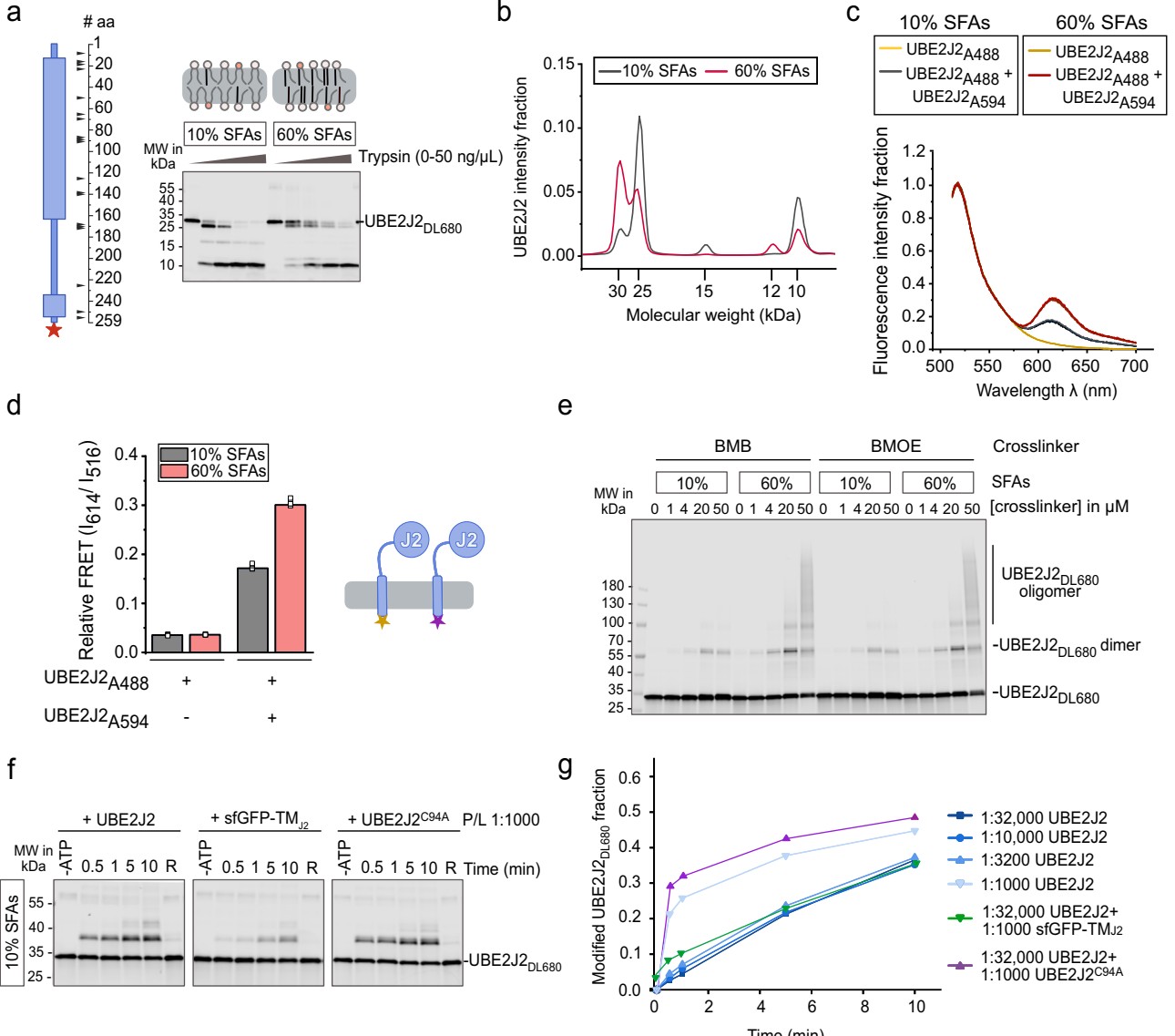

**Fig. 3 | Lipid packing affects UBE2J2 conformation. a** Limited proteolysis assay of UBE2J2$_{DL680}$ reconstituted into liposomes with indicated SFA content (P/L = 1:32,000), incubated with increasing trypsin concentrations (0, 1.9, 5.6, 16.6 or 50 ng/μl) at 25 °C for 30 min. Schematic (left) shows predicted tryptic sites (100% cleavage probability, PeptideCutter[102]; the red star indicates the position of the fluorescent label. Reaction products were analyzed by SDS-PAGE and fluorescence scanning. **b** Distribution of UBE2J2$_{DL680}$ fluorescence along the gel lane for proteoliposomes incubated with 1.85 ng/μl trypsin from (**a**). **c** Fluorescence emission spectra of liposomes reconstituted with the FRET donor UBE2J2$_{A488}$ in the absence or presence of the FRET acceptor UBE2J2$_{A594}$ (P/L = 1:20,000 and 1:5000, respectively). Each condition overlays six curves (two reconstitutions, each measured in triplicate). Data were normalized to the emission at 516 nm. **d** Relative FRET

efficiency calculated from spectra in (**c**) as the ratio of acceptor to donor emission intensity (I$_{614}$/I$_{516}$). **e** Cysteine crosslinking of UBE2J2$_{DL680}$ in liposomes with the indicated SFA content (P/L = 1:32,000), incubated with the indicated concentrations of 1,4-bismaleimidobutane (BMB) or 1,2-bismaleimidoethane (BMOE). Samples were analyzed by SDS-PAGE and fluorescence scanning. **f** Ubiquitin loading of UBE2J2$_{DL680}$, reconstituted in liposomes with 10% SFA content (P/L = 1:32,000), and the indicated proteins at different protein to lipid ratio (P/L). **g** Quantification of reactions in (**f**). For panels (**a**), (**e**), and (**f**), the experiments under these exact conditions were performed once; similar assays were repeated multiple times under related conditions and yielded consistent results. Detailed liposome compositions are given in Table 1. Source data are provided as a Source Data file.

C-terminal TMS and the disordered linker, confirming that most of UBE2J2 sits in the membrane with the correct orientation. However, at low trypsin concentrations, we observed differential exposure of tryptic sites. In loosely packed membranes (10% SFAs), the N-terminus was more susceptible to proteolysis, producing a ~25 kDa fragment, while it was less accessible in tightly packed membranes (60% SFAs, Fig. 3a, b). N-terminal tryptic site exposure correlated with low enzymatic activity. Similarly, in ER-like vesicles with 10 mol% cholesterol, in which UBE2J2 was inactive, the N-terminus was readily cleaved, whereas cleavage was subdued in POPL liposomes in which UBE2J2 was active (Supplementary Fig. 3a).

As a second indicator of conformational differences in different membrane environments, we assessed the propensity of UBE2J2 to dimerize or oligomerize. To this end, we measured fluorescence resonance energy transfer (FRET) between UBE2J2 molecules labeled at the C-terminus either with the donor fluorophore Alexa Fluor 488 (UBE2J2$_{A488}$) or with the acceptor fluorophore Alexa Fluor 594 (UBE2J2$_{A594}$). We found significantly higher FRET signal in membranes with 60% SFAs compared to those with 10% SFAs (Fig. 3c, d). Liposomes containing only donor or only acceptor showed minimal emission at 614 nm upon excitation at 480 nm, and low, lipid-independent FRET ratios (Fig. 3c, d and Supplementary Fig. 3b).

Thus, UBE2J2 TMS are more closely positioned in tightly packed membranes. This increased propensity for dimerization was confirmed by crosslinking assays. When UBE2J2 liposomes were incubated with the bifunctional cysteine crosslinkers 1,4-bis-maleimidobutane (BMB) or bis-maleimidoethane (BMOE), we observed more dimer- and oligomer-formation in 60% SFAs membranes than in 10% SFAs membranes (Fig. 3e).

Then, we tested the effect of increasing the local membrane density of UBE2J2 on its enzymatic activity. To this end, we co-reconstituted wild-type UBE2J2 with additional copies of either wild-type protein or UBE2J2 variants into the same liposomes, thereby raising the overall protein-to-lipid ratio (P/L) while keeping the total concentration of active enzyme in the reaction constant. Increasing P/L in this way alleviated the activity inhibition observed in unsaturated membranes (Fig. 3f, g). This response depended on the presence of the cytosolic region but not on catalytic activity as co-reconstitution with catalytically inactive UBE2J2 C94A relieved inhibition to a similar extent as wild-type. By contrast, co-reconstitution with superfolder GFP membrane-anchored via the UBE2J2 transmembrane helix (sfGFP-TMJ2) did not increase UBE2J2 loading, demonstrating that the effect cannot be explained by nonspecific membrane protein crowding.

At high P/L, UBE2J2 crosslinking increased and became less sensitive to SFA content (Supplementary Fig. 3c), indicating that dimerization correlates with high activity. However, dimerization does not appear to directly enhance catalytic activity or be strictly required, because soluble UBE2J2 lacking the TMS (sUBE2J2$_{1-226}$) exhibited rapid ubiquitin-loading kinetics but no evidence of dimerization (Supplementary Fig. 3d–f). We therefore propose that dimerization in the membrane context indirectly alleviates UBE2J2 inactivation by promoting a conformation similar to that induced by tight lipid packing. Supporting this notion, high-P/L reconstitutions reduced N-terminal accessibility even in loosely packed membranes, reproducing the proteolytic pattern seen in saturated membranes (Supplementary Fig. 3g). As dimeric UBE2J2 is unlikely to be the functional unit, we suggest that dynamic disassembly of dimers may precede E1 interaction and ubiquitin loading.

## UBE2J2 senses lipid packing via its cytosolic region

Next, we investigated the structural elements in UBE2J2 mediating its lipid sensitivity. Membrane anchoring was essential for this responsiveness, as soluble UBE2J2 was loaded rapidly with fluorescently labeled ubiquitin and was unaffected by the presence of liposomes (Supplementary Fig. 4a). To pinpoint regions involved in membrane sensing, we analyzed chimeric UBE2J2 variants in which the TMS and/or the disordered linker were replaced with corresponding regions from UBE2J1 (Supplementary Fig. 4a). Lipid saturation still activated these UBE2J2 variants (Fig. 4a, b). Similarly, fusing the cytoplasmic part of UBE2J2 to the TMS of synaptobrevin 2 (Syb2, Supplementary Fig. 4b, c) did not abolish lipid sensitivity. Conversely, UBE2J1 chimeras incorporating the TMS or linker of UBE2J2 did not acquire lipid sensitivity (Supplementary Fig. 4d–f). Thus, membrane anchoring is required, but neither the TMS nor the disordered linker of UBE2J2 contains specific lipid-sensing elements. We conclude that the UBC domain is primarily responsible for sensing lipid packing, though general physicochemical properties shared by the disordered linkers of UBE2J1 and UBE2J2 or between the tested TMS may also contribute.

We hypothesized that the cytosolic region of UBE2J2 associates with membranes exhibiting packing defects through hydrophobic interactions. To disrupt such interactions, we used the detergent *n*-octyl-β-D-glucoside (OG). At low concentrations, OG does not solubilize membranes (Supplementary Fig. 4g), but increases lipid packing by intercalating in the outer leaflet of proteoliposomes[54] (Supplementary Fig. 4h). In unsaturated membranes, low OG concentrations accelerated UBE2J2 ubiquitin loading (Fig. 4c, d), whereas activity in saturated membranes was largely insensitive to OG (Supplementary

Fig. 4i). This OG-induced activation correlated with a conformational change: limited proteolysis of loosely packed proteoliposomes (10% SFAs) treated with 2.6 mM OG (~7.5 times below the critical micellar concentration) showed a cleavage pattern similar to that seen in saturated membranes, whereas the cleavage pattern in tightly packed membranes was unaffected (Fig. 4e). Together, these observations support the existence of hydrophobic interactions between the cytoplasmic region of UBE2J2 and membranes with packing defects.

## Molecular dynamics simulations identify a potential membrane interacting region

To identify potential membrane-interacting regions in the UBC domain and better understand the conformational dynamics of UBE2J2 in the membrane, we performed molecular dynamics simulations of the membrane-embedded protein. The simulation system was designed to be large enough to accommodate the dynamics of the disordered linker region. Since no well-defined membrane interaction motif was discernible in the cytosolic region, protein-membrane interactions were expected to be unspecific and transient, posing a challenge for adequate sampling of conformational space. To address this, we used a coarse-grained representation of the system, using the Martini 3 force field[55].

UBE2J2 was simulated in ER-like membranes, or PC/PE membranes with either 10% or 60% SFAs. For each condition, 10 replicates were simulated for at least 20 μs. Comparison of the different membranes showed that reducing SFA content from 60% to 10% strongly increased the probability of encountering shallow lipid packing defects, while ER-like membranes (33% SFAs and 10% cholesterol) displayed an intermediate effect[56], indicating that the simulated membranes exhibited the expected physicochemical properties (Supplementary Fig. 5a). In all conditions, we observed reversible association of the UBC domain with the membrane. To quantify how lipid composition influences UBC domain association, we calculated the number of contacts between the UBC domain (residues 1–168) and the membrane. The contact distribution was qualitatively similar across all conditions, with two main peaks: one at zero contacts (unbound state) and another at ~30 contacts (bound state, Fig. 4f). To distinguish membrane-bound and -unbound populations, we modeled the contact data as a mixture of Poisson distributions. Across all conditions, a four-component model best fit the data, representing unbound, weakly bound (mean ~8 contacts), bound (~30 contacts), and tightly bound (~55 contacts) states. The population size of each conformation varied significantly with membrane composition.

Based on experimental observations, we expected that increased lipid packing would favor the unbound state. In the simulations, we also found that the ratio of bound to unbound conformations depended on membrane lipid composition (Supplementary Fig. 5b), albeit to a smaller extent than indicated by biochemical experiments. In ER-like and 10% SFA membranes, the UBC domain adopted the bound conformation 60% of the time, compared to 54% in 60% SFA membranes. To determine which residues mediate membrane binding, we calculated the lipid contact frequency for each UBE2J2 residue (Fig. 4g, h). Interactions were centered around a few hydrophobic residues on the UBC surface, particularly M1, M53, F100, and F137. Notably, these interactions clustered around the active site and N-terminus, rather than being uniformly distributed, suggesting a certain degree of specificity of the UBC domain–membrane interactions.

## F137E mutation desensitizes UBE2J2 to lipid packing

We focused our experimental validation of these candidate regions on aromatic amino acids, as their side chains have often been implicated in lipid sensing processes[16–18]. Strikingly, substitution of F137 with a negatively charged residue (UBE2J2$^{F137E}$) rendered UBE2J2 insensitive to lipid packing defects (Fig. 4i, j). In contrast, the F137Y mutant remained susceptible to inactivation by packing defects, similarly to the

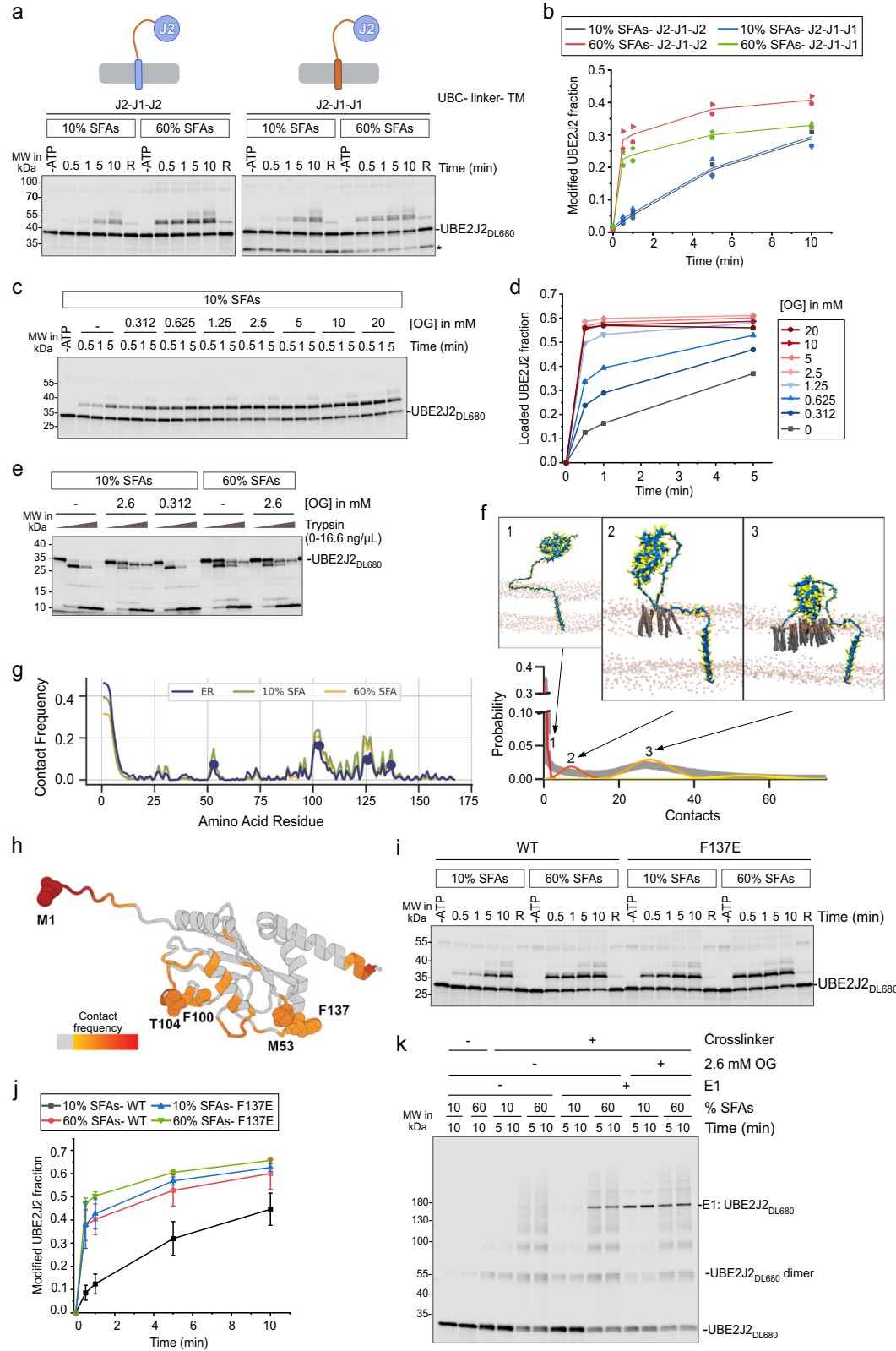

wild-type protein (Supplementary Fig. 5c). Intriguingly, when UBE2J2$^{F137E}$ was reconstituted in membranes exhibiting packing defects, it showed a tryptic digestion pattern reminiscent of wild type UBE2J2 reconstituted in highly saturated membranes; specifically, the N-terminus was less susceptible to proteolysis (Supplementary Fig. 5d and Fig. 3a). These results suggest that the F137E mutant adopts an active conformation independent of the membrane composition. In

contrast, mutation of another aromatic residue within the catalytic center, F100A, did not affect UBE2J2 inactivation in unsaturated membranes (Supplementary Fig. 5c).

We reasoned that association of the cytosolic region of UBE2J2 with the membrane impairs its interaction with E1, thereby inhibiting ubiquitin loading. To test this, we analyzed UBE2J2–E1 crosslinking using the cysteine crosslinker BMOE, and observed that crosslinking

**Fig. 4 | E1 interaction is impaired in membranes with lipid packing defects.**
**a** Ubiquitin loading of UBE2J2 chimeras with UBE2J1 TMS and/or disordered linker, reconstituted in PC/PE liposomes with the indicated saturated fatty acids (SFA) content. **b** Quantification of (**a**) (*n* = 2). *Co-purified UBE2J2 fragment. **c** Ubiquitin loading of UBE2J2$_{DL680}$ in liposomes with 10% SFAs, pre-incubated for 10 min with the indicated n-octyl-β-D-glucoside (OG) concentrations before loading. ATP was omitted where indicated (-ATP). **d** Quantification of (**c**). **e** Limited proteolysis assay on UBE2J2$_{DL680}$ proteoliposomes as in (**c**). Samples were pre-incubated with the indicated OG concentrations for 10 min before trypsinization (0, 1.9, 5.6 or 16.6 ng/μl) for 30 min at 37 °C. **f** Linegraph showing the probability of contacts between the UBC domain and the ER membrane from molecular dynamics simulations. The probability distribution can be described by a four component Poisson mixture model (unbound, loosely bound (with a mean of 8 contacts), and two tightly bound conformations (30 and 50 contacts, respectively)). Representative structures for the first three components are shown above. **g** Contact frequency

between the UBC domain and the different membranes (blue, ER-like; green, 10% SFAs; yellow, 60% SFAs). **h** Contact frequency for the ER-like membrane in (**g**) mapped onto the UBE2J2 structure. **i** Ubiquitin loading of wild type (WT) and mutant (F137E) UBE2J2, reconstituted PC/PE liposomes of indicated SFA content. **j** Quantification of (**i**) (*n* = 3, F137E; *n* = 7, WT 10% SFAs; *n* = 8, WT 60% SFAs). Error bars, mean ± sd. **k** Cysteine crosslinking to analyze UBE2J2:E1 interaction. UBE2J2$_{DL680}$ liposomes with indicated SFA content were incubated with the E1 ± 1,2-bismaleimidoethane (BMOE); in detergent-containing reactions, proteoliposomes were pre-incubated with OG for 10 min before crosslinker addition. Samples were collected at the indicated time points were analyzed by SDS-PAGE and fluorescence scanning. P/L = 1:32,000. In (**b**, **j**), each replicate represents an independent protein reconstitution into liposomes. For (**c**, **e**, and **k**), experiments under these exact conditions were performed once; similar assays were repeated multiple times under related conditions and yielded consistent results. Detailed liposome compositions are given in Table 1. Source data are provided as a Source Data file.

efficiency was much lower in unsaturated membranes compared to saturated ones (Fig. 4k, left). Notably, the same low OG concentrations that induced conformational changes in UBE2J2 also restored its interaction with E1 (Fig. 4k, right). Consistently, interaction of the lipid-insensitive UBE2J2$^{F137E}$ mutant with E1 was also enhanced (Supplementary Fig. 5e), in line with its observed activity and conformation in the membrane. Thus, UBE2J2 conformation defines its capacity to interact with E1 and undergo ubiquitin loading.

Collectively, these findings indicate that UBE2J2 adopts an auto-inhibitory conformation in membranes with packing defects. Hydrophobic interactions between its cytosolic region and the membrane hinder the recruitment of the E1 enzyme. In contrast, tight lipid packing disfavors these interactions, enabling efficient UBE2J2 loading.

## UBE2J2 activity instructs downstream reactions in the ubiquitination cascade

In the ER, UBE2J2 partners with several RING domain-containing E3 ligases. It promotes degradation of certain RNF139 substrates, including the tail-anchored protein HO-1, the misfolded cytosolic protein CL-1 and HLA class I molecules during viral immune evasion[57,58]. Loss of UBE2J2 increases steady-state levels of RNF139, indicating a role in RNF139 auto-ubiquitination[58]. Similarly, UBE2J2 collaborates with MARCHF6 to control the stability of MARCHF6 itself and its substrates, including the sterol biosynthetic enzyme SQLE[34,38,57]. To characterize E2-E3 collaboration biochemically, we purified and fluorescently labeled the E3 ubiquitin ligases RNF139, the homologous RNF145, and MARCHF6 for reconstitution into liposomes and also purified UBE2G2 and an engineered variant of its co-factor AUP1 (TM$_{Cue1}$-AUP1$_{271-410}$) (Supplementary Fig. 6a–h).

To analyze E3 activity in cooperation with UBE2J2 and UBE2G2, we measured E3 auto-ubiquitination. With UBE2J2 alone, all tested E3s were auto-ubiquitinated, confirming its collaboration with RNF139 and MARCHF6 and revealing a functional interaction with RNF145 (Supplementary Fig. 6i–k). Based on the molecular weight shifts, UBE2J2 primarily mediated multiple mono-ubiquitinations or the formation of short chains, while UBE2G2 generated longer, mainly K48-linked ubiquitin chains (Supplementary Fig. 6l), as reported before[59,60]. When both E2s were present, only long chains were observed, indicating that UBE2J2-initiated mono-ubiquitination is efficiently extended by UBE2G2. Thus, UBE2J2 acts as a priming E2, similar to Ubc6 in yeast[47,48,50]. In addition, most ubiquitin attachments on RNF139 were on serine or threonine residues, as indicated by the sensitivity to sodium hydroxide (NaOH)[61] (Supplementary Fig. 6m), corroborating that UBE2J2 mediates non-canonical ubiquitination[48].

Next, we tested whether the lipid-responsiveness of UBE2J2 affects downstream reactions in the ubiquitination cascade. We found that auto-ubiquitination of RNF145 was enhanced in membranes rich in saturated phospholipids (Fig. 5a, b) as was RNF145-mediated ubiquitination of UBE2J2 on non-cysteine residues (Fig. 5a, c, Supplementary

Fig. 7a). Similar lipid-dependent effects were seen for RNF139 auto-ubiquitination and MARCHF6-mediated UBE2J2 ubiquitination (Supplementary Fig. 7b–e).

Since all three ubiquitin ligases responded similarly to lipid saturation, these results suggest that UBE2J2 reactivity dictates E3 auto-ubiquitination. To determine whether membrane saturation also directly affects E3 ligase activity, we measured RNF145 auto-ubiquitination in the presence of soluble UBE2J2, whose loading is insensitive to membrane composition (Supplementary Figs. 3e and 4a). Under these conditions, RNF145 auto-ubiquitination was unaffected by lipid saturation (Fig. 5d). Furthermore, we assessed the lipid sensitivity of E3-mediated discharge of ubiquitin from UBE2G2 onto free ubiquitin. Proteoliposomes containing AUP1 and either RNF145, RNF139, or MARCHF6 were incubated with purified, ubiquitin-loaded UBE2G2 (Supplementary Fig. 7f) and free ubiquitin. The rate of di-ubiquitin formation was comparable across different lipid compositions for all E3s tested (Supplementary Fig. 7g). Together, these data indicate that membrane saturation affects E3 activity only indirectly, by modulating UBE2J2.

We next investigated whether lipid-responsiveness of UBE2J2, in concert with its E3 partner, impacts substrate ubiquitination. Since SQLE ubiquitination depends on UBE2J2 and MARCHF6, and is enhanced by lipid saturation in mammalian cells[26,27], we purified and reconstituted fluorescently labeled human SQLE into liposomes with varying SFA content (Supplementary Fig. 8a, b). MARCHF6- and UBE2J2-mediated SQLE ubiquitination was more efficient in membranes with 60% SFAs than in those with 10% SFAs, (Fig. 5e, f). This effect specifically required full length, membrane-anchored UBE2J2 (Fig. 5g), demonstrating that UBE2J2 governs the observed differences in SQLE ubiquitination. Under these experimental conditions, we did not detect any effect of saturation-induced conformational changes in SQLE on its ubiquitination independent of UBE2J2, contrary to previous hypotheses[26,62].

As UBE2J2 adopts different conformations depending on the membrane environment, we hypothesized that these dynamics not only affect interaction with E1, but also with E3 ligases. To assess this, we performed co-immunoprecipitation assays with proteoliposomes containing ALFA-tagged RNF145, UBE2J2, and Syb2 as a non-interacting control. After solubilization and precipitation of RNF145-ALFA, we found that significantly more UBE2J2 co-purified with RNF145-ALFA from membranes with 60% SFAs, compared to those with 10% SFAs (Fig. 5h, i). Syb2 did not co-purify under either condition, confirming the specificity of the assay. Co-purification required co-reconstitution of UBE2J2 and RNF145 within the same membrane, ruling out post-solubilization interactions (Supplementary Fig. 8c). Moreover, immunoprecipitation of intact liposomes without prior solubilization showed equal co-reconstitution efficiencies, indicating that differences in co-purification reflect membrane-induced changes in protein interactions (Fig. 5h, i).

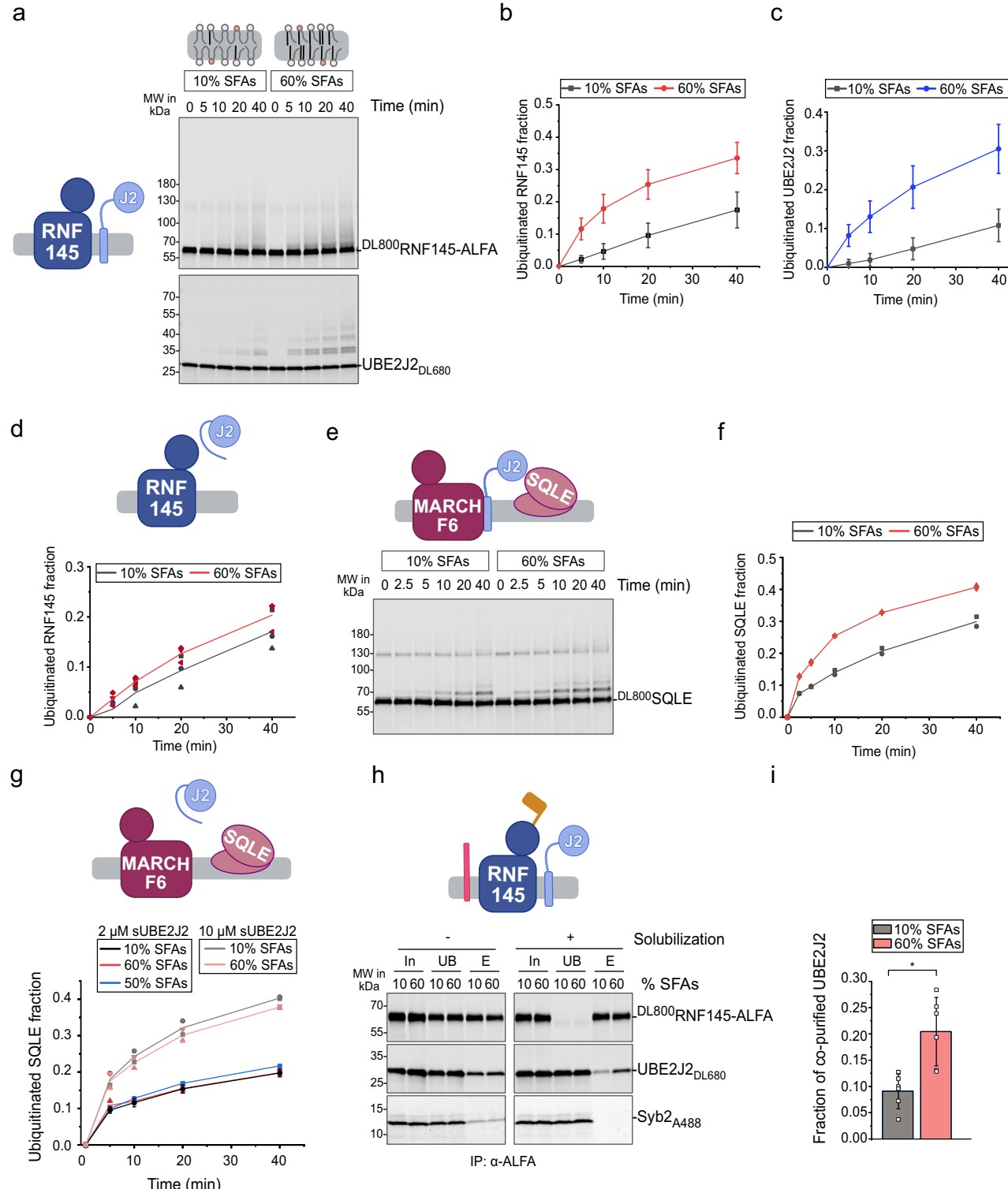

The influence of the membrane environment on UBE2J2–E3 interactions depended on protein concentration. An 8-fold increase in membrane protein concentration abolished the lipid sensitivity of UBE2J2–RNF145 binding (Supplementary Fig. 8d). Similarly, at higher protein levels, the effects of lipid saturation on RNF145 auto-ubiquitination and MARCHF6-mediated SQLE ubiquitination were diminished (Supplementary Fig. 8e–g). These results suggest a binding equilibrium between UBE2J2 and its cognate E3 ligase exists in both tightly and loosely packed membranes, and that high local E3 concentrations can overcome the inhibitory effect of packing defects on

E2-E3 interactions (Supplementary Fig. 8h). E3–bound UBE2J2 remains active even in less densely packed membranes, while packing defects otherwise stabilize an inactive UBE2J2 conformation. Thus, lipid saturation impacts not only UBE2J2 loading but also downstream reactions in the ubiquitination cascade.

### Cholesterol affects auto-regulation of the E3 ligase RNF145

Finally, we wanted to determine whether the aggregate activity of an E2/E3 pair can integrate multiple lipid signals – specifically, lipid saturation and cholesterol levels. Supplementing culture media with

**Fig. 5 | UBE2J2 activity modulates ubiquitination cascade outcome.**
**a** Ubiquitination of RNF145 and UBE2J2. Fluorescently labelled [DL800]RNF145-ALFA (P/L = 1:32,000) and UBE2J2[DL680] (P/L = 1:32,000) were co-reconstituted in PC/PE liposomes with the indicated saturated fatty acid (SFA) content. Samples collected at the indicated time points were analyzed by reducing SDS–PAGE and fluorescence scanning. Quantification of ubiquitinated RNF145 ((**b**), $n = 10$) and UBE2J2 ((**c**), $n = 7$ for 10%, $n = 6$ for 60% SFAs) from reactions as in (**a**). Solid lines connect means, error bars mean ± s.d. Each replicate is an independent protein reconstitution into liposomes. **d** Quantification of RNF145-ALFA ubiquitination with soluble UBE2J2 (sUBE2J2[1-226]). [DL800]RNF145-ALFA was reconstituted in liposomes with indicated SFA content (P/L = 1:32,000) and incubated with E1, ubiquitin, ATP and 0.5 μM sUBE2J2[1-226] ($n = 2$). **e** Ubiquitination of SQLE by MARCHF6 and UBE2J2. [DL800]SQLE, MARCHF6 and UBE2J2[DL680] were co-reconstituted in PC/PE liposomes with the indicated SFA content (each at P/L = 1:32,000). Samples were collected over time and analyzed as in (**a**). **f** Quantification of ubiquitinated SQLE from (**e**) ($n = 2$). **g** Quantification of SQLE ubiquitination by MARCHF6 and soluble UBE2J2.

[DL800]SQLE and MARCHF6 were co-reconstituted in PC/PE liposomes with the indicated SFA content (each at P/L = 1:16,000); sUBE2J2 was added at the indicated concentrations. **h** RNF145–UBE2J2 interaction analysis. [DL800]RNF145-ALFA, UBE2J2[DL680] and Syb2[A488] were co-reconstituted in liposomes with indicated SFA content (each at P/L = 1:32,000). Immunoprecipitation (IP) was performed with biotinylated anti-ALFA nanobody, eluted with SUMO protease Ulp1*. Co-reconstitution was estimated from IP of intact liposomes; protein-protein interactions were assessed by IP from detergent-solubilized liposomes. In: Input, UB, unbound, E elution **i** Quantification of UBE2J2[DL680] co-purifying with RNF145-ALFA from solubilized liposomes in (**h**). Co-purified UBE2J2[DL680] was normalized to eluted RNF145-ALFA and UBE2J2[DL680] input. Error bars, mean ± s.d. Statistical significance was assessed using a paired-sample, two-sided $t$-test (*$p = 0.00283$, $n = 6$). In (**b**, **c**, **d**, **f**, **g**, and **i**), each replicate represents an independent protein reconstitution into liposomes. Detailed liposome compositions are given in Table 1. Source data are provided as a Source Data file.

excess cholesterol reduces RNF145 and RNF139 steady-state levels through both transcriptional repression and increased proteasomal degradation[30,32,33] (reproduced in Supplementary Fig. 9a, b). For RNF145, this degradation requires its own E3 ligase activity[30]. To distinguish direct effects of cholesterol on RNF145 activity from indirect effects mediated through UBE2J2, we used soluble UBE2J2. Under these conditions, cholesterol enhanced RNF145 auto-ubiquitination in membranes with ER-like SFA levels (Fig. 6a, b). Likewise, cholesterol also increased RNF145 auto-ubiquitination when full-length UBE2J2 was present in membranes with 60% SFAs—a setting in which UBE2J2 ubiquitin loading is unimpeded by packing defects and not responsive to cholesterol (Figs. 6c, d, cf. and 2i). A weaker effect was observed when measuring RNF145 auto-ubiquitination in the presence of AUP1 and UBE2G2 (Supplementary Fig. 9c, d). Together, these results indicate that cholesterol directly promotes RNF145 activity, while lipid saturation regulates it indirectly, through UBE2J2.

Cholesterol did not directly modulate E3 activity of RNF145, as the discharge of ubiquitin from UBE2G2 by RNF145 was unaffected by the presence of cholesterol (Supplementary Fig. 9e). However, cholesterol increased RNF145 oligomerization, an effect independent of SFA content (Fig. 6e, f), which may enhance intermolecular (in-trans) auto-ubiquitination. In reactions containing both UBE2J2 and UBE2G2, cholesterol slightly increased polyubiquitination of a catalytically inactive RNF145 (C537, 540S, RNF145[CS]) mutant by active RNF145 (Supplementary Fig. 9f, g). In the absence of a catalytically active RNF145, some RNF145[CS] was modified by a mono-ubiquitin. Collectively, these results suggest that cholesterol may influence RNF145 auto-ubiquitination by promoting RNF145 self-association and in-trans auto-ubiquitination.

## Discussion

ERAD plays an important role in maintaining the identity of the ER membrane, defined by a specific proteome and lipidome. On the protein level, ERAD mediates degradation of proteins that are mistargeted in different ER domains and contact sites[63–66]. At the lipid level, it regulates the abundance of many lipid-metabolizing enzymes, thus influencing membrane composition[22,28,31,67]. Cellular studies have established that ERAD activity is responsive to changes in membrane lipid composition, but the complexity of the ER environment—with multiple membrane-embedded factors acting in parallel—has made it difficult to pinpoint the specific lipid-sensing components or to dissect their molecular mechanisms. To overcome this limitation, we used a fully reconstituted in vitro system with purified ERAD components and membranes of defined lipid composition, allowing us to isolate the contribution of individual factors to lipid responsiveness. In this study, we identify protein components of the ERAD ubiquitination cascade with membrane-sensing features and characterize their mechanisms of

signal integration. We show that membrane composition modulates the ubiquitination machinery at several stages.

At the E2 level, we identify UBE2J2 as a sensor for lipid packing (Fig. 7). In loosely packed membranes, UBE2J2 adopts conformations in which its cytosolic region associates with membrane packing defects, thereby preventing E1 recruitment and ubiquitin loading. This membrane-driven inactivation can be relieved by UBE2J2 dimerization, association with a cognate E3 ligase in membrane regions enriched in ERAD factors, or increased phospholipid saturation, all of which promote an active conformation. The existence of an equilibrium between active and membrane-inactivated UBE2J2 is inferred from measurements of UBE2J2 loading in an experimental protocol, where proteoliposomes are pre-equilibrated prior to E1 addition, allowing UBE2J2 to adopt either conformation before the ubiquitin loading reaction begins. This setup differs from the physiological context, in which E1 is continuously present and UBE2J2 cycles between loaded and discharged states. To determine whether membrane-driven inactivation of UBE2J2 significantly affects the overall ubiquitination process, it is necessary to consider the steady-state levels of ubiquitin-charged UBE2J2 under such continuous cycling conditions. As modeled in Supplementary Fig. 10a–c (and discussed in detail in the Supplementary Discussion), the steady-state fraction of Ub-loaded UBE2J2 is strongly influenced by the conformational equilibrium only when loading and discharge cycles are frequent, as expected in vivo. In such a dynamic setting, membrane composition can substantially impact the capacity for effective ubiquitination by controlling the pool of active UBE2J2.

At the E3 level, RNF145 has been proposed to be directly sensitive to cholesterol levels[30]. In agreement with this, we find that cholesterol enhances RNF145 auto-ubiquitination, independent of the E2 enzyme and lipid saturation, possibly by inducing RNF145 dimerization. This suggests that cholesterol could also affect the RNF145 interactome more broadly, as shown for its interaction with INSIG and subsequent HMGCR recruitment and ubiquitination[30,35]. The mechanism of cholesterol sensing by RNF145 remains unresolved. It may involve direct cholesterol binding to a putative SSD situated in its N-terminal transmembrane domain[30,35], although this SSD differs structurally from those in SCAP and HMGCR (Supplementary Fig. 10d, e). Alternatively, RNF145 might sense cholesterol levels indirectly, via the impact of cholesterol on global membrane properties such as membrane thickness, fluidity and packing. However, our data do not support a direct sensing of membrane fluidity as previously proposed[31]. Although increased lipid saturation raises RNF145 auto-ubiquitination, in agreement with its destabilization in cells[31], we propose this is due to enhanced UBE2J2 activity in saturated membranes. Consistent with this, palmitic acid treatment reduces UBE2J2 levels in cells, likely reflecting increased auto-ubiquitination.

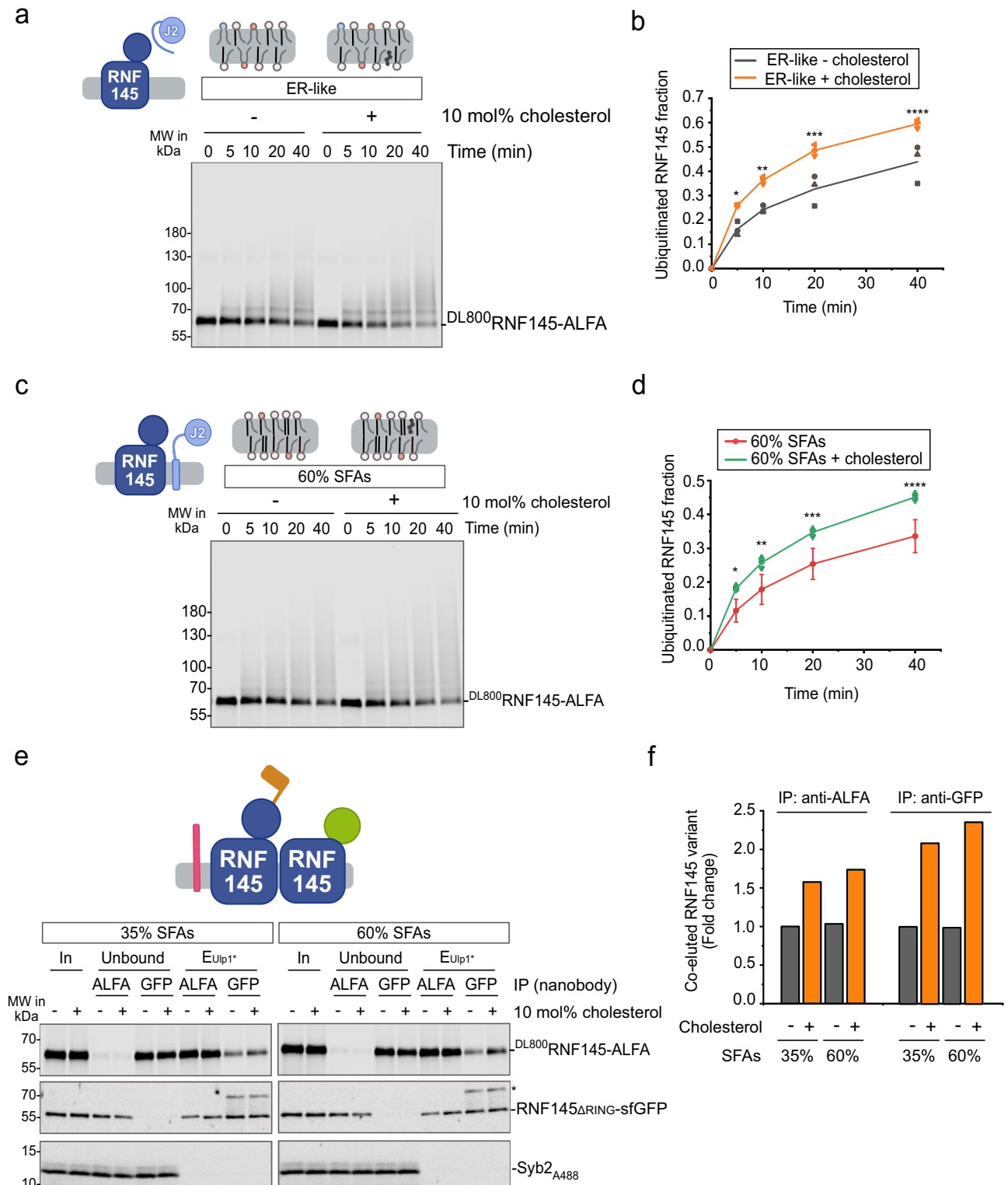

Our reconstituted system, while allowing detailed mechanistic analysis, lacks the full lipid complexity of native ER membranes. Excluded species such as phosphatidic acid, diacylglycerol, sphingolipids, and phospholipids containing polyunsaturated fatty acids, known to impact lipid packing[2,68–70], may also influence UBE2J2 activity. In addition, the exact distribution of proteins and lipids among individual liposomes is not directly controlled. Although we adjusted protein-to-lipid ratios and used defined lipid mixtures, stochastic variation between vesicles may generate local environments that differ from the average composition. Such variation could influence quantitative aspects of the reactions, such as local protein crowding, but is unlikely to affect the qualitative dependencies on lipid packing and cholesterol observed here. Furthermore, ER lipid composition is unlikely to be homogeneous across the entire organelle[71]. Such heterogeneity−including microdomains enriched, for example, in saturated lipids or pronounced curvature−may locally modulate UBE2J2 activity and enable ER subdomain-specific regulation of ERAD. Future studies will be necessary to determine how such lipid species

**Fig. 6 | Cholesterol modulates auto-regulation of the E3 ubiquitin ligase RNF145. a** Ubiquitination of RNF145 with a soluble UBE2J2 (sUBE2J2$_{1-226}$). Fluorescently labeled $^{DL800}$RNF145-ALFA was reconstituted in ER-like liposomes with or without 10 mol% cholesterol (P/L = 1:32,000) and was incubated with E1, ubiquitin, ATP and 2 μM sUBE2J2$_{1-226}$. Samples were analyzed by reducing SDS-PAGE and fluorescence scanning at indicated time points. **b** Quantification of ubiquitinated RNF145 from (**a**) (n = 3). Error bars indicate mean ± s.d. Statistical significance between conditions at each time point was assessed using Welch's two-sample, two-sided t-test: p = 0.02561 (5 min, *), p = 0.00057 (10 min, **), p = 0.03553 (20 min, ***), p = 0.06854 (40 min, n.s.). **c** Ubiquitination of RNF145 with UBE2J2. Proteins were co-reconstituted in liposomes with 60% SFA content, with or without 10 mol% cholesterol (P/L = 1:32,000 for both proteins). Samples were analyzed as above. **d** Quantification of ubiquitinated RNF145 from (**c**), n = 3 (with cholesterol), n = 10 (without cholesterol). Error bars indicate the mean ± s.d. Statistical significance at each time point was determined using Welch's two-sample, two-sided t-test:

p = 0.000134 (5 min, *), p = 0.00071 (10 min, **), p = 0.00011 (20 min, ***), p = 0.000024 (40 min, ****). **e** Analysis of RNF145 oligomerization. $^{DL800}$RNF145-ALFA was co-reconstituted with Syb2$_{A488}$ and RNF145$_{ΔRING}$-sfGFP (the C-terminal cytoplasmic part including the RING domain was replaced with superfolder GFP) in (liposomes of indicated SFA content ±10% cholesterol (P/L = 1:32,000). After solubilization, proteins were precipitated with anti-GFP or anti-ALFA nanobodies. Samples were analyzed by SDS-PAGE and fluorescence scanning. The experiment under these exact conditions was performed once; similar assays were repeated multiple times under related conditions and yielded consistent results. **f** Quantification of co-purified RNF145 upon immunoprecipitation of RNF145-ALFA or RNF145$_{ΔRING}$-sfGFP from the experiment in (**e**). The co-purifying RNF145 variant was normalized to nanobody-precipitated RNF145 and to the input. In (**b** and **d**), each replicate represents an independent protein reconstitution into liposomes. Detailed liposome compositions are given in Table 1. Source data are provided as a Source Data file.

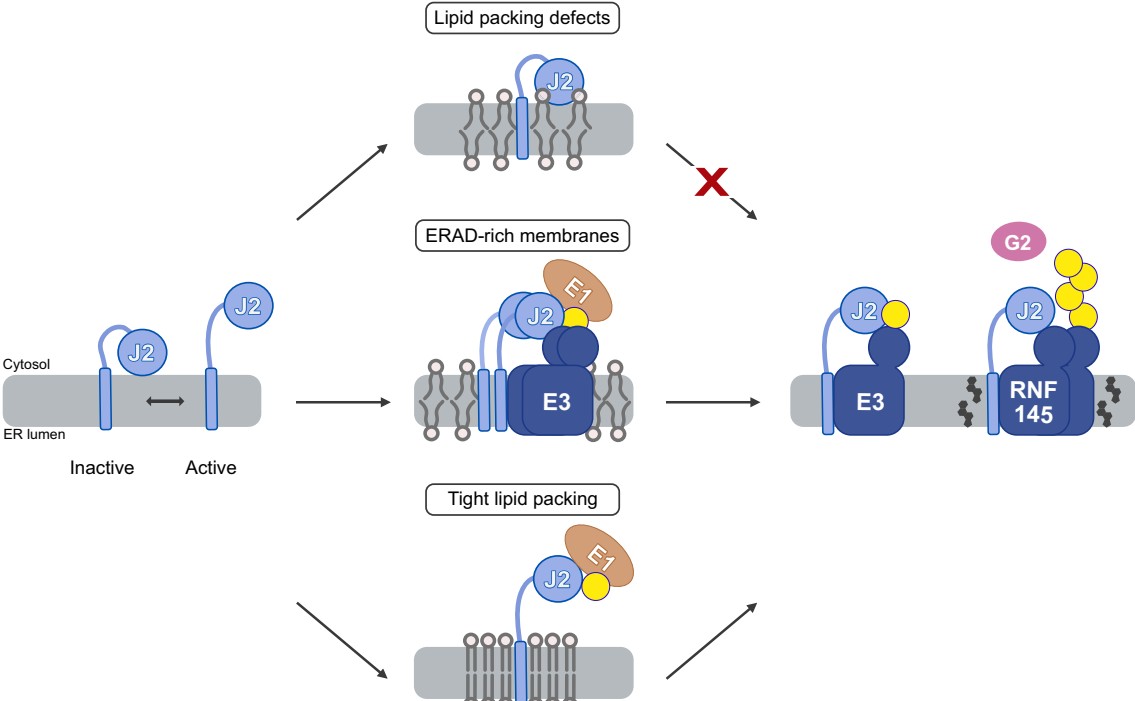

**Fig. 7 | Model for regulation of the ubiquitination cascade by membrane composition.** UBE2J2 adopts different conformations dependent on the membrane environment, thereby controlling its activity. Hydrophobic interactions between the cytosolic region of UBE2J2 and the membrane stabilize an auto-inhibited conformation. In the active state (left), the cytosolic region is accessible for E1-mediated loading. Membranes enriched in unsaturated lipids, exhibit packing defects (top), stabilizing an inactive UBE2J2 conformation that impairs interaction with the E1 and E3 enzymes. Increasing local UBE2J2 concentration overrides this inhibition via dimerization, promoting interaction with E1 and an E3 (middle). Increasing lipid saturation enhances lipid packing and favors an active UBE2J2 conformation. Ubiquitin loading of UBE2J2 and physical interaction with its cognate E3s, including RNF145, is enhanced (bottom). RNF145 is directly regulated by cholesterol, which might promote its oligomerization and auto-ubiquitination (right).

and ER subdomains together influence UBE2J2 function in the native context.

Our findings also reveal an unusual mechanism of membrane property sensing by UBE2J2. Unlike sensors such as Mga2, IRE1, and PERK[15,18], which sense lipid bilayer stress through their transmembrane domains, UBE2J2 senses packing defects through its cytosolic UBC domain. We identified specific surface-exposed hydrophobic residues in this region that mediate transient association with loosely packed membranes; mutation of one of these residues (F137) abolishes lipid sensitivity without impairing catalytic competence. This sensing mechanism is reminiscent of ALPS motifs or proteins like α-synuclein, which recognize packing defects via amphipathic helices[72], but is distinct in that UBE2J2 uses the surface features of a core enzymatic domain rather than a dedicated membrane-binding motif.

Consequently, autoinhibitory membrane association arises from subtle, reversible UBC–lipid interactions that respond to changes in bilayer packing.

Regulation of ERAD activity via UBE2J2 offers several potential advantages. Membrane-based inactivation of UBE2J2 could limit non-specific ubiquitination, whereas activation—either temporally (for example during ER stress) or spatially (in ER subdomains with tight lipid packing or high ERAD-component abundance[71,73,74])—could stimulate broader ERAD activity. Because UBE2J2 cooperates with multiple E3 ligases (MARCHF6, RNF139, RNF145), it can coordinate the turnover of diverse lipid-metabolic enzymes[23,30,31,67,75], as shown for SQLE in our reconstituted system. Its role as the priming E2 in ubiquitin chain initiation—often a rate-limiting step for chain formation[47,76–78]—means that modulating UBE2J2 can influence the overall throughput of

the cascade. Furthermore, its ability to ubiquitinate not only lysines, but also serines and threonines[24,39] expands the substrate repertoire of its partner E3 ligases when activated. Finally, additional direct regulation at the E3 level enables ERAD to integrate multiple signals, allowing for specific adjustment of activity toward distinct substrate sets defined by the E3 involved.

## Methods

### Molecular cloning
Plasmids for protein expression in *Escherichia coli* (*E. coli*) were cloned using commercially available kits for Gibson assembly and site directed mutagenesis (New England Biolabs, NEB). Plasmids for protein expression in *S. cerevisiae* were generated by GoldenGate assembly (NEB) based on[79]. Protein identifiers and expression plasmids are summarized in Supplementary Tables 4 and 5. Expression plasmids for RNF145, RNF139 and MARCHF6 were toxic for bacterial growth. Final assemblies were therefore directly transformed in yeast.

### Protein expression and membrane fractionation of *E. coli*
Protein expression in *E. coli* was performed using BL21-CodonPlus (DE3)-RIPL cells (Agilent). Media components are listed in Supplementary Table 2. Protein expression in Terrific Broth[80] was induced with 0.5 mM IPTG for 16 h at 18 °C, shaking at 130 rpm. Bacterial pellets were harvested by centrifugation at $4000 \times g$ for 20 min at 4 °C, and resuspended by homogenization in lysis buffer containing 50 mM Tris/HCl pH 7.8, 500 mM NaCl, 30 mM imidazole, 200 mM sucrose, 1 mM PMSF. Bacterial cells were mechanically lysed by three passages through a high-pressure microfluidizer at 16 bar. For soluble proteins, the lysate was cleared by ultracentrifugation at $185,000 \times g$ for 30 min at 4 °C. For membrane proteins, lysate was precleared by centrifugation at $4000 \times g$ for 15 min at 4 °C, then a crude membrane pellet was obtained by ultracentrifugation as above. The membrane pellet was washed once in lysis buffer and the ultracentrifugation step was repeated. Membranes were stored in lysis buffer at −80 °C.

### Protein purification from *E. coli*
Membrane fractions were solubilized for 1 h rotating at 4 °C in purification buffer (50 mM Tris/HCl pH 7.8, 500 mM NaCl, 30 mM imidazole, 200 mM sucrose) containing 1% (w/v) detergent and 1 mM PMSF. Solubilized proteins were separated by insoluble material by ultracentrifugation as above. Affinity purification was performed using HisPur Ni-NTA agarose resin (Thermo) and elution was performed by incubation of protein-bound beads with 1 μM Ulp1 for 30 min at 4 °C.

**UBE2J2, UBE2J1 and variants.** Membrane fractions of UBE2J2 variants were solubilized in 1% (w/v) DDM. After affinity purification, proteins were fluorescently labeled and further purified by size exclusion, using a Superdex 200 Increase 10/300 GL column equilibrated in 20 mM HEPES/NaOH pH 7.4, 400 mM NaCl, 200 mM Sucrose, 0.5 mM TCEP containing 0.03% (w/v) DDM.

**Soluble UBE2J2 and UBE2G2.** After affinity purification, proteins were further purified by size exclusion, using a HiLoad 16/600 Superdex 75 pg column, equilibrated in 20 mM HEPES/NaOH pH 7.4, 200 mM NaCl, 200 mM Sucrose, 0.5 mM TCEP.

**AUP1.** Membrane fraction containing $TM_{Cue1}$-$AUP1_{271-410}$ was solubilized in 1% DM. After affinity purification, AUP1 was further purified by size exclusion, using a Superdex 200 Increase 10/300 GL column equilibrated in 20 mM HEPES/NaOH pH 7.4, 300 mM NaCl, 200 mM Sucrose, 0.5 mM TCEP containing 0.2% (w/v) DM.

### Protein expression and membrane fractionation of *S. cerevisiae*
For protein expression in yeast a *ubc7* deletion strain derived from BY4742 (OpenBiosystems) with genotype *MATα ura3Δ0 his3Δ1 leu2Δ0 lys2Δ0 ubc7::kanR* was used[81]. The general expression protocol using galactose induction was used as described[81]. The E1 enzyme Uba1 was expressed in the InvSc1 strain (Invitrogen). Yeast pellets were harvested at $4000 \times g$ for 20 min at 4 °C, washed once in water, snap-frozen in liquid nitrogen as beads and stored at −80 °C.

Lysis of frozen yeast was performed mechanically by cryo-milling, using the Planetary Ball Mill PM 100 (Retsch). Yeast beads were subjected to 10 grinding cycles of 2.5 min, including 1 min of reverse rotation, at 400 rpm. Yeast powder was homogenized in membrane fractionation buffer (20 mM HEPES/NaOH pH 7.4, 400 mM NaCl, 200 mM Sucrose, 1 mM PMSF and 2 μM pepstatin A). Whole cell lysate underwent an ultracentrifugation round at $104,350 \times g$ for 20 min at 4 °C and the pellet containing unbroken cells and cell membranes was again homogenized in membrane fractionation buffer. Lysis was further assisted by a single passage through microfluidizer (M-110L, Microfluidics) at a pressure of 6 bar. The unbroken cells and cellular debris were pelleted by centrifugation at $2500 \times g$ for 10 min at 4 °C. The supernatant containing cell membranes was subject to ultracentrifugation at $204,526 \times g$ for 45 min at 4 °C. The pellet containing the membrane fraction was washed and stored in fractionation buffer at −80 °C.

### Protein purification from *S. cerevisiae*
Membrane fractions were solubilized for 1 h rotating at 4 °C in purification buffer (20 mM HEPES/NaOH pH 7.4, 400 mM NaCl, 200 mM Sucrose, 0.5 mM TCEP) containing detergent and 1 mM PMSF. Solubilized proteins were separated from insoluble material by ultracentrifugation at $204,526 \times g$ for 30 min at 4 °C. Proteins expressed in *S. cerevisiae* harbored a streptavidin binding protein (SBP) purification tag[82], at the N- or C-terminus. Affinity chromatography for all proteins was performed using streptavidin agarose resin (SERVA) and elution was performed with 2 mM biotin.

**RNF145, RNF139 and variants.** Proteins were expressed with an N-terminal SBP tag followed by a $SUMO_{EuB}$ protease cleavage site[83] and a C-terminal ALFA tag[84]. Membrane fractions containing RNF145-ALFA, $RNF145_{CS}$-ALFA, $RNF145_{\Delta RING}$-ALFA, $RNF145_{\Delta RING}$-sfGFP and RNF139-ALFA were solubilized in 1.3% (w/v) GDN. After elution, the SBP-$SUMO_{EuB}$ tag was cleaved off by incubating with 1 μM $Senp_{EuB}$ protease for at least 1 h at 4 °C. Proteins were further purified by size exclusion using a Superose 6 Increase 10/300 GL column in buffer containing 20 mM HEPES/NaOH pH 7.4, 250 mM NaCl, 200 mM Sucrose, 0.5 mM TCEP, 0.006% (w/v) GDN.

**MARCHF6.** MARCHF6 was expressed with an N-terminal SBP tag followed by a $SUMO_{EuB}$ protease cleavage site[83] or C-terminal SBP preceded by a TEV cleavage site and followed by a sortase-labeling sequence. Membrane fractions were solubilized in buffer containing 1% (w/v) DMNG, which was reduced to 0.012% (w/v) during affinity purification. Eluted protein was further purified by size exclusion using a Superose 6 Increase 10/300 GL column in buffer containing 20 mM HEPES/NaOH pH 7.4, 150 mM NaCl, 0.2 mM TCEP, 0.012% (w/v) DMNG.

**SQLE.** SQLE was expressed with an N-terminal SBP tag followed by a $SUMO_{EuB}$ protease cleavage site[83]. Membrane fractions were solubilized in buffer containing 1% (w/v) DM, which was reduced to 0.2% (w/v) during affinity purification. Eluted protein was further purified by size exclusion using a Superdex200 Increase 10/300 GL column in buffer containing 20 mM HEPES/NaOH pH 7.4, 250 mM NaCl, 0.2 mM TCEP, 0.2% (w/v) DM. Uba1 was purified, $Syb2_{A488}$ purified and fluorescently labeled as described in ref.[81].

### Fluorescent labeling of purified proteins
Purified proteins were covalently modified with a fluorescent dye via a transpeptidation reaction catalyzed by the bacterial Sortase A which

mediates the ligation of G-rich sequences at the carboxyl terminus of LPxTG sequences[85]. For N-terminal labeling, recombinant proteins harbored a GGG sequence at their N-terminus, which was exposed upon proteolytic cleavage of the affinity purification tag and was linked to fluorophore-conjugated CLPETGG peptide. For C-terminal fluorescent labeling, recombinant proteins contained a C-terminal LPETGG sequence which was linked to fluorophore-conjugated GGC peptide. In the labeling reaction, purified proteins were incubated with the fluorophore-conjugated peptide (1:2.5 molar ratio), 2 μM sortase A and 5 mM CaCl$_2$ for 2 or 16 h at 4 °C. Free peptide and sortase were removed gel filtration using Superose 6 Increase 10/300 GL or Superdex200 Increase 10/300 GL columns equilibrated to the same buffers as used before.

### Preparation of liposomes

Lipids and detergents used for liposome preparation are listed in Supplementary Table 1. Large unilamellar vesicles were generated by reverse phase evaporation[86]. In detail, synthetic lipids were mixed at defined molar ratios to a final concentration of 20 mM in chloroform. A dry lipid film was generated by chloroform evaporation at 100 mbar for 5 min and at 20 mbar for 30 min. Subsequently, lipids were dissolved in 1 mL diethyl-ether, followed by addition of 300 μl liposome buffer (20 mM HEPES/KOH pH 7.4, 150 mM KCl, 5 mM magnesium acetate). A lipid emulsion was generated by tip sonication. Diethyl ether was removed by evaporation at 500 mbar for 5 min, until a gel-like phase was formed. Then, 700 μl liposome buffer was added to a final volume of 1 mL. Liposomes were formed with sequential, 1 h-long cycles of diethyl ether evaporation at 300 mbar, 200 mbar and 100 mbar. Uniform liposome size was achieved by filtered extrusion through a 0.4 μm and 0.1 μm poly-carbonate filters (Mini extruder kit, Avanti Polar Lipids). Detailed lipid compositions are described in Table 1.

### Protein reconstitution in liposomes

Proteoliposomes were generated as described[50]. Briefly, 4 mM of liposomes were partially solubilized by addition of DM to a final concentration 8.2 mM in a total volume of 100 or 130 μl reconstitution buffer (20 mM HEPES/KOH pH 7.4, 150 mM KCl, 5 mM magnesium acetate, 0.2 mM TCEP). Purified proteins were added at defined protein to lipid ratios (P/L), and the mixture was incubated for 1 h at ambient temperature. Detergent was removed in three 20-min-long incubation cycles with 350 μg/μL Pierce detergent removal resin (DRR) (Thermo Scientific) at ambient temperature. After each cycle, DRR was removed by centrifugation at 2500 × g for 2 min using Pierce micro-spin columns (Thermo Scientific).

### Flotation assay

Proteo-liposomes were placed at the bottom layer of a density gradient containing 40% (100 μl), 30% (40 μl), 15% (40 μl) and 0% (40 μl) (w/v) Nycodenz in reconstitution buffer (20 mM HEPES/KOH pH 7.4, 150 mM KCl, 5 mM magnesium acetate, 0.1 mM TCEP). The density gradient was subjected to ultracentrifugation at 258,826 × g for 90 min at 4 °C in a S55-S rotor, during which reconstituted proteins co-migrated with lipids at the top, least dense density fractions. Six fractions (36.5 μL each) were collected from the top of the density gradient and analyzed by SDS-PAGE and fluorescence (Li-Cor) or stain-free (BioRad) scanning.

### Antibody accessibility assay

Antibody accessibility assays were performed as described previously[50] using a Tecan Genios Pro microplate reader. First, the basal fluorescence of Alexa Fluor 488-labeled E2 (f.c. 100 nM) in liposomes or detergent was measured using 495/10 nm and 535/25 nm for excitation and emission, respectively. Subsequently, anti-Alexa Fluor 488 polyclonal antibody (A-11094, Invitrogen) was added at a dilution of 1:15 and changes in fluorescence were measured over time. After 10 min, liposomes were solubilized by addition of 1% Triton X-100 (1:20 dilution from a 20% stock) and further reduction in fluorescence was measured until signal was stabilized. The data of these three individual measurements were merged, background fluorescence after detergent addition was subtracted, and data were normalized to maximal basal fluorescence.

### Limited proteolysis and protease protection assay

Protease protection assays were used to analyze protein orientation and conformation. Proteoliposomes were incubated with the TEV protease, Ulp1 or trypsin (Roche) at concentrations, temperature and time courses indicated below each experiment. Proteolysis was stopped by addition of 5 mM PefaBloc (Roche) and incubation at ambient temperature for 10 min. Subsequently, 3x LDS sample buffer (12% lithium dodecyl sulfate (w/v), 30% glycerol (w/v), 0.05% Coomassie blue G-250 (Serva), 150 mM Tris/HCl (pH 7.0)) was added, protein samples were heated at 50 °C for 10 min and analyzed by SDS-PAGE and fluorescence scanning or western blotting.

### Dynamic light scattering

The average diameter of (proteo)-liposomes was determined by dynamic light scattering (DynaPro Titan, Wyatt Technology) using the Dynamics V6 software. In brief, (proteo)-liposomes were diluted to a final concentration of 1 mM lipid into buffer containing 20 mM HEPES/KOH pH 7.4, 150 mM KCl, 5 mM magnesium acetate. Each measurement took place at 25 °C, with 3% laser power, including 10 cycles of 10 s each.

### Table 1 | Compositions of liposomes used in this study

| Lipid species (mol%) | Lipid mix (SFA content) | | | | | | | | | | |
|---|---|---|---|---|---|---|---|---|---|---|---|
| | 33% SFA (ER-like) | 10% SFA | 35% SFA | 40% SFA | 50% SFA | 60% SFA | 35% SFA +cholesterol | 60% SFA +cholesterol | 10% SFA (PC only) | 60% SFA (PC only) | 10% SFA (16 carbons) |
| DOPC | | 60 | 10 | | | | 6.7 | | 80 | | |
| 16:1 PC | | | | | | | | | | | 60 |
| POPC | 60 | 20 | 70 | 80 | 80 | 60 | 66.7 | 56.7 | 20 | 80 | 20 |
| DPPC | | | | | | 20 | 16.7 | 16.7 | | 20 | |
| DOPE | 20 | 20 | 20 | 20 | | | | | | | |
| 16:1 PE | | | | | | | | | | | 20 |
| POPE | | | | | 20 | 20 | | 16.7 | | | |
| DOPS | 10 | | | | | | | | | | |
| Cholesterol | 10 | | | | | | 10 | 10 | | | |

*SFA* saturated fatty acid.

## C-Laurdan measurements

C-Laurdan measurements were performed as described, with minor modifications[18]. Briefly, 0.5 mM liposomes were incubated with 0.5 μM C-Laurdan (R&D systems) in liposome buffer (20 mM HEPES/KOH pH 7.4, 150 mM KCl, 5 mM magnesium acetate) for 30 min at 37 °C. Recording of the fluorescence emission spectrum was carried out in a plate reader (BioTek Synergy Neo 2) at 37 °C (excitation=375 nm (10 nm range), emission spectrum =400–600 nm (20 nm range) with a 2 nm wavelength step). Generalized polarization (GP) was calculated based on the equation below, where $I_{p1}$ equals the sum of intensity at wavelength range 402–460 nm and $I_{p2}$ equals the sum of intensity at wavelength range 470–530 nm.

$$GP = \frac{Ip1 - Ip2}{Ip1 + Ip2}, \quad -1 < GP < 1 \tag{1}$$

## Ubiquitin loading assay

Ubiquitin loading assays were performed in ubiquitination buffer (20 mM HEPES/KOH pH 7.4, 150 mM KCl, 5 mM magnesium acetate, 0.1 mM TCEP). Proteoliposomes containing 50 nM E2 enzyme were incubated at 37 °C for 10 min prior to addition of the ubiquitination machinery, containing yeast Uba1 at a concentration of 1 μM for UBE2J2 or 4 μM for UBE2J1, 50 μM yeast ubiquitin, 2.5 mM ATP and 0.2 mg/mL BSA. At different timepoints, samples were transferred into 3x LDS sample buffer with or without 6% (v/v) β-mercaptoethanol and were analyzed by SDS-PAGE and fluorescence scanning.

## Ubiquitination assay

Ubiquitination assays were performed in ubiquitination buffer (20 mM HEPES/KOH pH 7.4, 150 mM KCl, 5 mM magnesium acetate, 0.1 mM TCEP). Proteoliposomes containing 50 nM E3 ligase were mixed with the ubiquitination machinery, including 0.1 μM yeast Uba1, 50 μM yeast ubiquitin, 0.2 mg/mL BSA on ice. Reactions were placed at 37 °C and ubiquitination started by addition of 2.5 mM ATP. At different timepoints, samples were transferred into 3x LDS sample buffer with or without 6% (v/v) β-mercaptoethanol and analyzed by SDS-PAGE and fluorescence scanning.

## Quenched-flow and kinetic models

Rapid quenched kinetic analysis was carried out using an RQF-3 quench flow instrument (KinTek Corporation). Equal volumes of reaction solution A, containing 100 nM UBE2J2$_{DL680}$ in liposomes, and reaction solution B, containing 2 μM Uba1, 100 μM Ubiquitin and 5 mM ATP, both in ubiquitin loading buffer and equilibrated to 37 °C, were rapidly mixed. Reactions were quenched using 0.1 M HCl and collected in 3x LDS sample buffer-containing tubes. Samples were analyzed by SDS-PAGE and fluorescence scanning. For the analysis data from Fig. 2a, b was included for t ≥ 30s and ≤ 10min.

For kinetic fitting, we assumed a simple mechanism, in which an active and inactive conformation of UBE2J2 (A and B, respectively) interconvert with rates $k_1$ and $k_2$, respectively, and only the active conformation can react with $k_3$ with ubiquitin-loaded E1 to yield C, ubiquitin-loaded UBE2J2, yielding the following set of equations:

$$\frac{d[C]}{dt} = k_3[A] \tag{2}$$

$$\frac{d[A]}{dt} = -k_3[A] - k_1[A] + k_2[B] \tag{3}$$

$$\frac{d[B]}{dt} = k_1[A] - k_2[B] \tag{4}$$

In this model, the starting concentrations $[A]_O$ and $[B]_O$ are determined by

$$\frac{[B]_0}{[A]_0} = \frac{k_1}{k_2} \tag{5}$$

and the total concentration of loadable UBE2J2, $[T]$, defined as $[T] = [A]_O + [B]_O$. This set of equations was solved using a formalism described by Andraos[87], resulting in

$$C(t) = T\left[\frac{k_3 k_2}{k_1 + k_2}\frac{e^{r_1 t} - e^{r_2 t}}{r_1 - r_2} + \frac{k_3 k_1}{k_1 + k_2}\frac{\frac{e^{r_1 t} - 1}{r_1} - \frac{e^{r_2 t} - 1}{r_2}}{r_1 - r_2}\right] \tag{6}$$

with

$$r_{1,2} = \frac{-(k_1 + k_2 + k_3) \pm \sqrt{(k_1 + k_2 + k_3)^2 - 4k_2 k_3}}{2} \tag{7}$$

Global fitting was then performed using the Origin software (Version 2025, OriginLab Corporation, Northampton, MA, USA), in which $k_3$ was shared, concentrations expressed as fractions of the total UBE2J2, and $T$ fixed as 0.75. Discharge of UBE2J2 by hydrolysis of the thioester was neglected in this model.

## Discharge assays

For discharge of UBE2J2, UBE2J2$_{DL680}$ was loaded with ubiquitin, as described above. Subsequently, ubiquitin loading reactions were quenched by dilution (1:10) into EDTA-containing buffer (20 mM HEPES/KOH pH 7.4, 150 mM KCl, 2 mM EDTA, 0.1 mM TCEP), and discharge was monitored over time.

For discharge of UBE2G2, UBE2G2 was loaded with fluorescently labeled ubiquitin in a reaction containing 73 μM UBE2G2, 3 μM Uba1, 120 μM fluorescent ubiquitin and 2 mM ATP in ubiquitination buffer (20 mM HEPES/KOH pH 7.4, 150 mM KCl, 5 mM magnesium acetate, 0.1 mM TCEP). After incubation at 37 °C for 100 min, reaction was stopped by addition of 10 mM EDTA, and ubiquitin-loaded UBE2G2 was purified by size exclusion in buffer containing 10 mM MES pH 6.0, 150 mM NaCl, 0.5 mM TCEP. Proteo-liposomes (f.c. 100 nM of E3 ligase) were mixed with 100 μM ubiquitin in buffer containing 20 mM HEPES/KOH pH 7.4, 150 mM KCl, 5 mM magnesium acetate, 0.1 mM TCEP and were pre-equilibrated at 37 °C for 5 min. Subsequently, 100 nM ubiquitin-loaded UBE2G2 was added and discharge was monitored over time. Samples were transferred in 3x LDS sample buffer and analyzed by SDS-PAGE and fluorescence scanning.

## Proteoliposome immunoprecipitation assay

Immunoprecipitations took place in buffer containing 20 mM HEPES/KOH pH 7.4, 150 mM KCl, 5 mM magnesium acetate, 0.1 mM TCEP. 10 μl Pierce magnetic streptavidin beads (Thermo) were coated with 2 μM biotinylated anti-ALFA or anti-GFP nanobody in a total volume of 20 μl. Subsequently, free streptavidin was blocked with 0.1 mM biotin-PEG. Nanobody-coated beads were washed and incubated with blocking solution containing 2 mg/mL BSA for 5 min to prevent non-specific interactions. Proteoliposomes used as input were either intact or pre-solubilized with detergent for 30 min at ambient temperature. Proteoliposome input containing 0.2 mg/mL BSA was incubated with nanobody-coated beads (bead to input ratio of 1:2) and incubated for 1 h at ambient temperature. Beads were washed three times with pull-down buffer containing 0.2 mg/mL BSA (and 0.5% (w/v) DM for solubilized liposomes only). Elution was performed with 1 μM Ulp1* to release nanobody-protein complex from coated beads in DM-containing buffer. Samples were analyzed by SDS-PAGE and fluorescence scanning.

 

## FRET assays

Liposomes were reconstituted with donor or acceptor fluorophore-labeled protein at defined protein to lipid ratios. Proteo-liposomes were incubated for 30 min at 37 °C in liposome buffer 20 mM HEPES/KOH pH 7.4, 150 mM KCl, 5 mM magnesium acetate, 0.1 mM TCEP containing 0.2 mg/mL BSA. Samples were excited with a wavelength of 480 nm and fluorescence emission spectra were recorded in a plate reader (BioTek Synergy Neo 2) at 37 °C.

## Cysteine crosslinking

Proteo-liposomes (f.c. 100 nM protein) were incubated with the crosslinker bis(maleimido)ethane (BMOE) or 1,4-bis(maleimido) butane (BMB) (Thermo Scientific) at 25 °C for 30 or 45 min in buffer containing 20 mM HEPES/KOH pH 7.4, 150 mM KCl, 5 mM magnesium acetate. Crosslinkers were quenched by addition of 10 mM DTT for 10 min at ambient temperature. Then, protein samples were mixed with 3x LDS sample buffer and analyzed by SDS-PAGE and fluorescence scanning.

## Protein analysis by SDS-PAGE

SDS-PAGE was performed based on the classical method described by Laemmli (1970). Proteins mixed with LDS buffer[88] were heated at 50 °C for 5 min and loaded onto a 4–20% Mini-PROTEAN® TGX™ or Criterion™ TGX Precast Gel (Bio-Rad). The marker used was the PageRuler™ Plus Prestained Protein Ladder, 10 to 180 kDa (Thermo Scientific). Electrophoresis was performed in 1x Laemmli buffer (3 g/L Tris, 14.4 g/L glycine, 1 g/L SDS). Proteins were visualized using a Gel Doc™ EZ system for stain-free detections or by staining using Instant Blue Coomassie stain (0.008% (w/v) G250 Coomassie Blue G-250, 9.4% (w/v) orthophosphoric acid (85%), 0.25% (w/v) each cyclodextrin (α-, β-, γ-). Fluorescence scanning was performed in an Odyssey DLx imager (Licor) or in a Typhoon™ FLA 7000 (GE).

## Western blot

Proteins were transferred from the gel onto a nitrocellulose membrane using the Trans-Blot® Turbo™ semi-dry transfer system (Bio-Rad) according to the manufacturer's guidelines. Nitrocellulose membranes were incubated in blocking solution (5% (w/v) skim-milk powder in Tris-buffered saline containing 20 mM Tris-HCl pH 7.4, 150 mM NaCl, 0.1% (v/v) Tween20 (TBS-T) for 1 h at ambient temperature. Subsequently, they were incubated with the primary antibody solution (αSBP 1:1000 (MAB10764); αRNF145 1:1000 (HPA036562); αGAPDH (MA515738) 1:2000 in 5% (w/v) skim-milk powder in TBS-T) for 1 h at ambient temperature or overnight at 4 °C. After three 5-min washes in TBS-T, membranes were incubated with the IRDye conjugated secondary antibody (1:15,000, Li-Cor) for 1 h shaking at ambient temperature. Washing steps were repeated and fluorescence was detected using an Odyssey scanner (Li-Cor).

## Image analysis and quantifications

Quantification of immunoprecipitation assays and western blots was carried out in ImageStudio™ (Li-Cor). Quantification of ubiquitin loading and ubiquitination was performed in FIJI[89]. The fraction of modified protein was calculated by normalization of modified band (or lane area) intensity to total lane intensity. Graphs were prepared using the Origin software (Version 2018b, OriginLab Corporation, Northampton, MA, USA).

## Mammalian cell culture

HeLa cells were kindly provided by Dr. Sonja Lorenz, MPI-NAT. HeLa cells were cultured in Dulbecco's modified Eagle's high glucose medium (DMEM) supplemented with 10% fetal bovine serum (FBS) and 50 units/ml penicillin-streptomycin. Cultures were maintained in 10 cm plastic culture dishes at 37 °C and 5% $CO_2$. Media components are listed in Supplementary Table 3.

Manipulation of cholesterol content was performed according to available protocols[30]. Briefly, HeLa cells were cultured in complete medium (DMEM containing 5% (v/v) FBS, penicillin, streptomycin) until they reached 70% confluency (~30 h). Subsequently, complete medium was removed and cells were washed three times with pre-warmed PBS. For sterol depletion, HeLa cells were then cultured in sterol depletion medium (DMEM containing 5% (v/v) LPDS, 50 µM mevalonate, 10 µM mevastatin) for ~20 h. Subsequently, cholesterol was replenished by supplementing the medium with 50 µM cholesterol complexed with β-methyl-cyclodextrin (chol:β-MCD). For preparation of mevalonate, mevalonolactone was hydrolyzed under alkaline conditions by incubation with 2.04 M KOH. The chol:β-MCD stock was prepared based on published protocols[90]. First, powdered cholesterol was dissolved in chloroform: methanol solution (1:1) at a final concentration 50 mg/mL. The solvent was removed by rotary evaporation at 100 mbar pressure for 5 min and 20 mbar for 30 min. The dry lipid film was dissolved by tip sonication in 25 mM β-MCD in water to a final cholesterol concentration of 2.5 mM. The solution was incubated overnight at 37 °C under rotation, filtered and stored at −20 °C.

## HeLa cell lysis and total protein extraction

Cells growing on dishes were washed twice with ice cold PBS and were scraped off into PBS on ice. Cell pellets were obtained by centrifugation at $2500 \times g$ for 3 min using a benchtop centrifuge at 4 °C and resuspended in lysis buffer (50 mM Tris/HCl pH 7.5, 150 mM NaCl, 1 mM EDTA, 0.1% (w/v) SDS, 1% (v/v) Triton-X100) supplemented with 1 mM PMSF and protease inhibitor cocktail. After incubation on ice for 40 min, unbroken cells and debris were removed by centrifugation at $17,000 \times g$ for 15 min. Samples were analyzed by SDS-PAGE and western blotting.

## Molecular dynamics simulations

**Coarse-grained model of full length UBE2J2 embedded in model membranes.** Initial coordinates of UBE2J2 were taken from the x-ray structure 2F4W[91]. The linker and transmembrane helix were modeled based on the Alphafold2 prediction[92]. The structure was centered on the transmembrane helix and rotated so that the helix was aligned with the z-axis. Subsequently we used Martinize2, distributed with version 0.13.0 of the vermouth library to generate the coarse-grained structure and protein topology[93]. The Martini3-IDP force field was used to assign force field parameters[55,94]. Elastic net restrains were applied to the E2 domain (residues 13–170) and transmembrane helix (residues 229–259)[95]. Pairwise distance restraints were applied based on an upper distance cutoff of and 0.9 nm, a lower cutoff distance of 0.5 nm and a force constant of 700 kJ/mol/nm². These parameters best reproduced the root mean squared fluctuation of E2 domain in reference all-atom simulations. After generating the initial distance restraints, we removed the restraints on the flexible active site loops (residue 99–103 and 126–129). The protein was subsequently embedded in the model membrane systems using a modified version of the insane2 script. The updated Martini3 lipid parameters and Martini3 cholesterol parameters were used to parameterize the membrane[96,97]. Solvent molecules and potassium chloride beads were added. The ion concentration was 150 mM, additional ions were added to neutralize the charge of the system. The Martini3 parameters were used for the water and ion beads, for potassium we used the SQ4 bead recommended in the original Martini3 publication[55].

The simulation systems were then subjected to a two-stage energy minimization. First, position restraints were applied to the protein (500 kJ/mol/nm²) and the phosphate beads of the lipids (100 kJ/mol/nm²). In the second stage, the system underwent energy minimization without position restraints. Both energy minimizations used the steepest descent algorithm and were carried out to machine precision. After energy minimization, the systems underwent a series of short

simulations. The Verlet scheme was used in all simulations. Pairwise interactions were evaluated explicitly up to 1.2 nm. The potential shift method was used for van der Waals (vdW) interactions to shift them to zero at the cutoff, and the reaction field method was used for long-range electrostatic interactions with a dielectric constant of 15. In the first simulation, the systems were simulated for 200 ps, maintaining a constant system volume. An integration timestep of 2 fs was used. The system was coupled to a Berendsen thermostat with a coupling constant of 1 ps and a target temperature of 298 K[98]. Initial velocities were assigned from a Maxwell distribution. In subsequent equilibration simulations, initial velocities were assigned from the final frame of the previous simulations. Position restraints were applied to the backbone and lipid phosphate beads with force constants of 1000 and 200 kJ/mol/nm², respectively. The system was simulated for 400 ps in the second simulation. The position restraints were from the previous simulation. The system was coupled to a Berendsen barostat with a target pressure of 1 bar and a coupling constant of 3 ps. Pressure coupling was implemented independently in the $xy$ plane and $z$ direction (semi-isotropic). This simulation was followed by a second NPT simulation in which the force constants of the position restraints were reduced to 500 and 100 kJ/mol/nm². The integration time step was increased to 10 fs, and the system was simulated for 25 ns. For the final equilibration simulation, we removed the position restraints fully and increased the integration timestep to 20 fs. We then simulated the system for 100 ns. Additionally, we switched from the Berendsen thermostat and barostat to the C-Rescale barostat and V-Rescale thermostat[99,100].

**Simulations.** The same parameters were used in the production runs as in the final equilibration simulation. Ten replicates were simulated, with initial velocities assigned randomly from a Maxwell distribution. Each replicate was simulated for at least 20 μs.

**Analysis.** Most analysis utilized tool implemented in GROMACS version 25.1 and custom python scripts build upon MDAnalysis version 2.70[101]. Additionally, we used PackMem to calculate membrane defects[56]. Packing defects were evaluated on a regular grid with 1Å grid spacing. Following the recommended protocol, we used the vdW radius of the coarse-grained beads to determine grid occupancy. Since this value is not provided directly in the Martini3 force field, it was calculated from the self-interactions. Statistical analysis of the packing defect was performed using python.

### Reporting summary
Further information on research design is available in the Nature Portfolio Reporting Summary linked to this article.

## Data availability
Simulation protocols, analysis pipeline and trajectories have been deposited to Zenodo (https://doi.org/10.5281/zenodo.17095023). The following PDB accession codes have been used in this study: 2F4W, 8DJM, and 7ETW. Source data are provided with this paper. MD Source data are provided with this paper.

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

## Acknowledgements

We thank Iris Bickmeyer for excellent technical assistance, Reinhard Jahn, Alex Faesen and Tayfun Hazar Eyyuboğlu for comments on the manuscript, and Sonja Lorenz, Ivo Feussner and Blanche Schwappach for discussions. Furthermore, we thank Marina Rodnina for providing access to a quenched-flow apparatus, and Olaf Geintzer for helping with measurements. We also thank the Facility for Light Microscopy at the MPI-NAT and Peter Lenart for technical support. This work was supported by the Deutsche Forschungsgemeinschaft SFB1190, P15 and by the European Research Council under the Horizon2020 research and innovation program (grant # 677770) (both to AS).

## Author contributions

A.V. carried out and analyzed all biochemical experiments. F.L. carried out and analyzed molecular dynamic simulations, C.S. established the protocols for MARCHF6 purification and contributed to the conceptual planning of revision experiments. A.S. conceived and supervised this work. H.G. supervised the molecular dynamic simulations. A.V. and A.S. wrote the original draft of the manuscript, all authors contributed to revision and editing of the manuscript.

## Funding

## Competing interests

The authors declare no competing interests.
