## [Transparent Peer Review file · Nature Communications]

UBE2J2 sensitizes the ERAD ubiquitination cascade to changes in membrane lipid saturation

Corresponding Author: Dr Alexander Stein

Version 0:

Reviewer comments:

Reviewer #1

(Remarks to the Author)

Endoplasmic reticulum-associated protein degradation (ERAD) are pathways of the ubiquitin-proteasome system responsible for identifying, ubiquitylating, and degrading misfolded or surplus proteins in the ER lumen and membrane. This process is essential for maintaining protein homeostasis within the ER and plays a role in regulating membrane properties by modulating the degradation of different lipid metabolic enzymes. ERAD's functionality relies on a coordinated cascade of enzymes, including E1 (ubiquitin-activating), E2 (ubiquitin-conjugating), and E3 (ubiquitin-ligase) enzymes. Recent findings suggest that ERAD's activity is influenced by the lipid composition of ER membranes; however, the mechanisms of how ERAD components sense and respond to such changes are not fully understood.

In this work, Vrentzou et al. examined the influence of membrane composition on the ubiquitylation cascade in ERAD. The team focused on E2 and E3 enzymes, particularly UBE2J2 and RNF145, reconstituting them in membranes with different lipid compositions. They reported that the E2 enzyme UBE2J2 can act as a sensor for lipid packing; its activity is impeded in loosely packed membranes, whereas tightly packed lipid environments facilitate its interaction with the E1 enzyme. Additionally, they reported that the E3 ligase RNF145 appeared to be cholesterol levels, which impacts its oligomerization and enzymatic activity. Through these biochemical studies, the authors attempt to provide new insights into how lipid signals are integrated by the ERAD machinery to maintain membrane homeostasis, implicating UBE2J2 as a central regulator for lipid-responsive ERAD activity.

However, I think this manuscript falls quite short of this apparent goal and over-claims throughout. The purifications and reconstitutions of different enzymes is quite elegant, but in many cases, the experiments are without the appropriate controls to ensure that the apparent difference is actually related to activity of the enzymes. The activity changes reported for UBE2J2 could all be explained by simple clustering, and in fact this is what my interpretation would be and their data would tend to agree. Regardless, in all cases the effects on enzymatic activity are quite small, with very dramatic lipid changes that are unlikely to ever occur, in situ.

Major concerns

Throughout the entire manuscript, there are no attempts to test any of the authors' conclusions within cells. For this reviewer, this calls the premise and conclusions into question. This fact, coupled with the unphysiological lipid compositions, make the story incomplete.

The lipid compositions used appear to be extreme conditions that do not appear in normal cellular situations. The small, inconsistently-sized functional changes of UBE2J2 loading caused by these extreme lipid changes make the authors' interpretations difficult to agree with.

All of these effects are quite small. Technical differences could really for all of these effects. This would likely be very difficult, but I can't think of a better way to be sure than the authors should develop mutants that are non-responsive to either SFAs (for E2) or cholesterol (for E3) to ensure that these are meaningful differences.

The authors should show determine and show the orientation of the E2 in key experiments to rule out orientation as a complicating factor.

How do the authors know that they have evenly distributed proteins between their liposomes? This would greatly affect their interpretations so it should be addressed experimentally (specifically in Figure 3D).

Similarly, the results in all cases with UBE2J2 simply point towards increased association, which could be driven by the uneven distribution of lipids within, or between, the liposomes. The crosslinking experiments would suggest this to be true.

If the authors' interpretation of Figure 2D is correct, the lipid packing would be increasing as increase OG, based on the reference cited. They should first use laurdan to test whether this is true, then, they should use their FRET assay from Figure 3D to test their interpretation.

The title does not fit, this is exaggerating the results described in this work. There is no description of How the enzymes do this task, only an indication that the activity changes. Similarly, the abstract misrepresents which is actually observed ("master regulator"). Broadly, I think some restraint should be used to describe the work accurately.

Why was UBE2J1 ignored? In many cases, UBE2J1 appears to be responsive (Extended Data 2D) similarly to UBE2J2. Broadly, whether experimental results are described as significant, or not, depends on whether the results are what the authors want to claim.

Minor concerns

Could small variations in the activity of the enzyme be contributing to some of these differences? I think this is reasonable because there are pretty large changes of the same conditions in different figures.

Immunoprecipitations from a reconstituted system are a bit unconventional. Some additional specificity controls are probably required.

Broadly, the authors are not particularly clear with the composition of the lipid bilayer. All of these results are interpreted as differences in saturation, but oleic fatty acids and palmitoyl fatty acids also differ in chain length. The lipid compositions used fail to account for potential differences in having chains distributed between lipids, for example 50% DOPC 50% DPPC is likely to behave differently compared to 100% POPC.

Why are all proteins so large on SEC? For example, UBE2J2 is 37kDa by SDS-PAGE, broad peak 400+ kDa by SEC

Other suggestions

Increased labeling of Figure 3A would help readers interpreting the results.

The intro focuses quite a bit on lipid properties and sensing within yeast at the transcription factor level, which is okay. But the authors have missed more recent related work on lipid influencing ERAD function, for example PMID 36638172.

Reviewer #2

(Remarks to the Author)

Vrentzou et al. report an important finding: E2 ligase UBE2J2 is the primary component of the ERAD response and is triggered by lipid packing density. This is a beautiful manuscript, featuring highlight-level biochemistry and crystal-clear conclusions. It also can be improved significantly on the data analysis side with relatively limited extra work (and a few additional experiments). I agree with the conclusion that the switch-like change in UBE2J2 is very likely a response to general membrane properties and not to individual lipids (as underlined by the assays in the presence of PE and cholesterol). But I also think that the data show a different regulatory mechanism in response to PE and Chol levels, which could be discussed a bit more in depth.

I am strongly in favour of publishing the paper in Nat. Commun. and recommend that the manuscript should be accepted after a few (but important) revisions.

Major point:

The authors obtain beautiful kinetics for E2 ligase loading in reconstitution assays that are far from easy. The level of biochemical ability to do these types of experiments with the time resolution that is presented here is rare indeed and should be applauded. However: The authors then proceed to simply do a linear interpolation of the data points, which is a severe under-utilization of the obtained rather fantastic data (please excuse the formulation, I want to get my point across). I strongly recommend that the authors fit a few simple kinetic models to their data (given that the plots are made with Origin anyway, this just means copy-pasting the integrated rate law and setting the boundary conditions for the fit), which should allow them (a) to underpin the conclusions of their impressive manuscript with quantitative kinetic analysis and (b) potentially uncover new mechanistic details.

Given that this is at saturating levels of ATP/Ub/E1 ligase, we're looking at a simple reaction which has a straightforward rate law – given the concentrations applied here it should simply be a bimolecular reaction, likely with first-order kinetics due to the high excess of reactants. By eye-balling the data, I'd say that that the corresponding rate law in its simplest form can be fitted only to one dataset: UBE2J2 after detergent treatment of the liposomes in Fig. 1. Interestingly, UBE2J1 looks to be

loaded with similar kinetics (the difference is thus unlikely a question of activity), but either a large fraction of the protein remains inaccessible or loading is an equilibrium reaction/there is a significant hydrolysis rate. This can be tested by depleting ATP from the reaction mixtures after reaching the steady state and following the loaded fraction over time with the corresponding kinetic analysis. I recommend doing this experiment for both proteins.

In all liposome datasets, the loaded fraction levels of around 0.5 which makes sense if the orientation of protein incorporation is roughly 1:1. However, I think (as do the authors judging from their formulation of “immediately available fraction for loading”) that the obtained kinetics for conditions of higher content of saturated PL cannot be explained by a simple bimolecular reaction model any more (Fig 2). The data look like two independent loading reactions (a very fast one and a slow one) overlaid as a linear combination – a clear indication that one cannot assume a well-mixed situation. Such a situation makes it difficult to draw any firm conclusion without fitting quantitative kinetic models, further underlying the need for this type of analysis.

Intriguingly, the detergent data in Fig 1C do not share this characteristic – they very much look like a single exponential could be fitted reasonably well. Thus, it is unlikely a feature of the protein itself. My guess: This is due to heterogeneity in the liposome lipid composition. A mixed initial PL composition will lead to a liposome population with a certain distribution of individual PL compositions, a fraction of which will be suitable to trigger the switch to UBE2J2 hyperactivity. Since it is unclear how addition of DPPC to a DOPE/DOPC system affects this distribution w/o having the data in hand I would probably not stress the 50% // 60% saturated side chain point overly much – there may well be nonlinear effects in play. The best model to fit here would be to assume two UBE2J2 pools of distinct activity that do not have any exchange with each other on the timescale of the experiment.

The following experiments are optional, but I do believe that they would further underline & illustrate the effects of lipid composition on enzyme activity:

The distribution of lipid packing in liposomes can be experimentally tested using e.g. the single particle profiler approach that Erdinc Sezgin recently published (<https://www.nature.com/articles/s41587-023-01825-5>), which allows to measure the GP of many individual liposomes. An experiment done with proteins in GUVs at relatively low magnification (20x) with fluorescent Ub would probably lead to some GUVs lighting up immediately, while others become fluorescent only over time.

Minor points:

- Intro: The lower SL content of the ER should also be mentioned in the characterization of ER membrane properties
- To me the UBE2J1 data look like there is a small but significant hyperactive fraction present under all conditions – it may well be the same conformational switch, but triggered by a different membrane property / something else.
- Line 155, should this read “PE increases packing”? The effects measured by GP and in the enzymatic assay point in different directions, if I read the data correctly. PE containing liposomes have a higher GP (higher packing) and also faster loading kinetics. Importantly, this effect is distinct from the effect of lipid saturation – it seems that PE inclusion accelerates the kinetics of the slow loading process. I am a bit confused. Please clarify
- Cholesterol seems to have a quirky effect – it accelerates the slow loading reaction, but also lowers the initial jump.
- Please plot quantification of the data in Fig 3F/G in a time resolved fashion as in Fig. 2. Looking at the gel I do not actually think that the effect disappears. It would be rather surprising if it did. I am not at all sure that the conclusion in this section holds up, but I am sure that fitting the two models discussed above will give a yes/no answer
- It would have been good to include the 6s timepoint in the experiment with the UBE2J2/Syb2 chimera in Fig 4a.
- Same for Fig 4e. Data are convincing, but would have been even more so with additional time resolution.
- I think that the detergent addition experiment does not conclusively address the packing defects point. That would be more convincingly done by showing the recruitment of a fluorescent biosensor for packing defects (e.g. a fluorescently labelled ALPS motive)
- With regard to the dimerization model of activation, the authors should check whether the faster loading kinetics cannot simply be explained by the higher enzyme concentration and amend Figure 7 and the discussion if necessary.

André Nadler

Reviewer #3

(Remarks to the Author)

Overview

In this manuscript, the authors investigate the membrane sensing properties within components of the mammalian ERAD mechanism using a reconstituted ubiquitylation system comprised of purified E3 and E2 enzymes embedded in proteoliposomes of varying composition. In doing so, the authors reveal several different levels at which membrane composition influences the ubiquitylation cascade. Most notably, they provide evidence that indicates the E2 enzyme UBE2J2 contains a region/domain within its cytoplasmically exposed portion, that effectively senses lipid packing – where loosely packed lipids containing low fraction of saturated fatty acids (10%) appeared to impede loading of activated ubiquitin onto the E2 UBE2J2 by the E1. When investigating the impact of phospholipid saturation, the authors found that an environment of tight packing (50-60% SFA) in contrast, permitted engagement and ubiquitin transfer by the E1. The loss of activity arising from slight adjustments to the SFA composition (40%>30%) would seem to indicate a degree of sensitivity with UBE2J2. Adding cholesterol slightly increased activity when SFA content was lower (35%) but did not add anything when it was higher (60%). The responsiveness to different compositions are used to support a hypothesis that UBE2J2 is a sensor for lipid packing. Limited proteolysis of UBE2J2 under different lipid compositions and release of an N-terminal fragment indicates a conformational change taking place. Evidence from TMD swapping is provided that sensing of lipid packing occurs through the cytosolic domain and is attributable to interaction/s between hydrophobic domain/s of UBE2J2 and

membranes with “packing defects”. As UBE2J2 works together with ERAD E3s that regulate sterol/lipids (e.g. MARCHF6, RNF139, RNF15), the authors suggest that this built in regulation helps to control membrane composition and that UBE2J2 sits at the centre as a “master regulator”. Evidence is provided that each E3’s auto-ubiquitylation was lipid dependent and attributable to UBE2J2, just as interaction with the E1 appears be. Collectively, the authors make a case for UBE2J2 as key component in the regulation of ER lipid composition through intrinsic features sensitive to local changes that impact the ubiquitylation cascade.

Overall impression

Mechanistic insight into membrane embedded ubiquitylation machinery participating in ERAD through reconstitution is a welcome addition to the plethora of genetic and cell-based data that has defined our understanding for some time. It should be of general interest both in its strategy and in its findings. This manuscript does a very good job of executing this technical study and in doing so, reaching some interesting conclusions about the potential interdependency of ubiquitylation machinery and ER membrane composition. It represents an advancement in the field as it highlights a potentially novel point of feedback on membrane composition. Overall, the manuscript is well written, the technical experiments performed to a high standard, the data of good quality. In most cases the appropriate statistical analyses have been included where necessary and are described. There were however several instances where the replicates were not sufficient (e.g. n=2). For the most part, the Methods and Materials describe the experiments that have been performed.

The main issues this reviewer has with the manuscript arise in two areas – 1) the certainty for which the term “master regulator” is used for UBE2J2 when the supporting evidence only partially addresses such a broad role and 2) the absence of any details regarding the “hydrophobic” interaction governing UBE2J2 and loosely/defectively packed lipids. For the former, while the impact changing lipid composition/packing has on UBE2J2 is clear, what is not clear is whether such a reductionist approach is sufficient to permit the use of the term “master regulator”, when for all intents and purposes, it’s nothing more than a conformational change that permits (or restricts) accessibility. Inclusion of some cell-based assays when lacking UBE2J2 and/or combined with treatments biasing ER lipid composition would go some way to earning the term “master regulator”. Regarding the latter, there is some concern within this reviewer that the domain/region of UBE2J2 that is purportedly acting to sense the membrane is not described in detail. The authors demonstrate the TMD (~aa 227-247) is not responsible. If not the UBC domain of UBE2J2 comprising the first ~160aa at the N-terminus, this leaves approximately ~60aa where a “sensor” might reside. Constructing chimeras with UBE2J1 could offer a straightforward approach to identifying the residues/domain involved. Alternatively, providing a Alpha Fold model highlighting hydrophobic patches within UBE2J2 (that maybe are not there in UBE2J1) could also help to support the authors conclusions.

Nevertheless, the manuscript is an advancement of understanding and a new vantage point for the how membrane composition can impact ERAD. There are several comments/suggestions that this reviewer has regarding the data presented. They are listed below.

Queries & Comments

1. For Figure 1B, Could the authors provide an explanation for the multiple bands observed with UBE2J2DL680 in detergent? There is some mention of loading and auto-ubiquitylation in the legend of Extended Data Fig 1D but this is not discussed anywhere else in the text and it is not clear why E1 transfer of Ub would produce such a pattern.
2. For Figure 1D, there is no corresponding reducing (R) lane for UBE2J1DL680 as there is for 1B. Was this an oversight?
3. In the text, the authors often refer to ER-like membranes. Could the authors speak to the heterogeneity of lipid composition within the ER membrane, it’s uniformity (or lack there of) within cells and speculate how this might impact localization of ubiquitylation machinery in it. More specifically, are the E2 and E3 enzymes partitioning in selected parts of the ER as a way to control/limit or engage their activity? Could the authors speculate whether it is the “active” or “inactive” state that UBE2J2 would preferentially/routinely reside in?
4. In Figure 3 and on pg 7, line 187-188 – Is there any in vivo or physiological evidence that UBE2J2 forms a dimer/oligomer? Moreover, while increasing the protein concentration of UBE2J2 appears to favour its activation, could this not just be attributable to non-specific crowding? Could the authors for example titrate in a catalytically dead UBE2J1 to drive home the point that UBE2J2 multimerization may important?
5. What domains of UBE2J2 are driving/influencing dimerization? Is it the same region responsible for sensing lipid packing? UBE2J1 chimeras might find some use here.
6. The TMD of UBE2J2 was swapped with that of Syt and sensitivity to lipid saturation was still observed. While demonstrating that sensitivity was not attributable to the sequence itself, UBE2J2 and Syt TMDs both have predicted hydrophobicities on the higher side (of the Kyte-Doolittle scale) for TA proteins. If a TMD from an ER-resident TA protein with lower hydrophobicity were to be swapped (like CYB5A or SQS), would the expectation still be that what senses packing was independent of membrane anchoring?
7. For Figure 5, is auto-ubiquitylation of the E3 an valid readout for a physiological process? Would it be technically feasible for a “substrate” to be introduced as well and ubiquitylation of it be used to assess the ubiquitylation cascade?
8. In Figure 5D an line 297, the authors state that the soluble UBE2J2 without a TMD was insensitive to membrane saturation. Yet the authors infer that the cytosolic portion of UBE2J2 that is hydrophobic is what senses lipid packing/saturation. How can this be resolved? Is membrane tethering a pre-requisite if not a determinant of UBE2J2 activation?
9. Regarding the reducing conditions of the samples, in the figure legends it is stated as 2% (v/v) 2-mercaptoethanol (R) although in the M&M it is written 6%. Which one is correct?

Typos and Text inconsistencies

- L38: membrane protein biogenesis 3-6, lipid biosynthesis

- L186: in which Ube2J2 was active (Extended Data Figure 3A).
- L376: thus loading of Ube2J2 with ubiquitin.
- L431: *S. cerevisiae*
- L437: need space after 66
- L438: 4000 x g
- L442: lysate was cleared by
- L443: 4000 x g, which temperature?
- L450: (w/v) detergent which one?
- L470: missing .
- L471: remove were
- L480: through a microfluidizer (?)
- L488: (w/v) detergent which one?
- L545: ultracentrifugation at 55000 rpm all the others are given in x g
- L565: 5 mM MgAc₂) for 30 minutes
- L566: 5 mM MgAc₂)
- L578 & L586: with or without 6% (v/v) β-mercaptoethanol the Fig. legends, it is written 2%
- L667 and L671: β-MCD
- L675: 2500xg
- L677: NaCl,
- L686 (table 5): pET39 His14-SUMO-UBE2J1S184D-TMUBE2J2-LPETGG

- Figure 1: Panel B and D: front size different for ER-like and detergent.
 Panel C and E: Loaded vs modified?
 L2: Ube2J1 and Ube2J2 both possess an N-terminal

- Figure 2: L4: deviation.(C) As in (A), (missing space)
 L9: ..described in 14.(F) Ubiquitin loading (missing space)

- Figure 4: Panel C and D: [OG] in mM vs [OG] (mM)
 L3: acids (SFAs).(B) (missing space)

- Ext Data Fig 2: L6: scanning.(C) Ubiquitin...(missing space)
 L14: described in 14.(H) Dynamic... (missing space)

Reviewer #4

(Remarks to the Author)

"I co-reviewed this manuscript with one of the reviewers who provided the listed reports. This is part of the Nature Communications initiative to facilitate training in peer review and to provide appropriate recognition for Early Career Researchers who co-review manuscripts."

Version 1:

Reviewer comments:

Reviewer #1

(Remarks to the Author)

In revision, the authors have significantly strengthened their manuscript and have addressed my major concerns- where possible with important data.

I appreciate the restraint from overclaiming in the rewriting of the manuscript, this has made the story more accurate. I think the authors should try to include some of the context from the questions raised by the reviewers in the discussion that were unable to be addressed. I don't think it takes away from the importance of the work, it just provides some more clarity for readers. For example around some uncertainty with protein and lipid distributions between liposomes.

But, I have no further substantial concerns and I believe this work should be published.

Reviewer #2

(Remarks to the Author)

The authors sufficiently addressed my points and I support the publication of the revised manuscript
 André Nadler

Reviewer #3

(Remarks to the Author)

In this revised manuscript, the authors have addressed most of this reviewers concerns. 2 minor points.

1. In line 72, the authors state that "UBE2J2 and UBE2J1 also modify serine and threonine residues thus expanding the substrate repertoire" and cite 2 papers to support this statement. While valid for UBE2J2, neither of these papers contain data demonstrating UBE2J1 has this capacity. While it might be anticipated, no formal evidence has been published. Perhaps this sentence could be reworked to reflect this subtle distinction.

2. Our initial review queried a line in Figure 1, Line 2.... "Ube2J1 and Ube2J2 both possess an N-terminal, catalytic..." which the authors were confused about and stated in their rebuttal. To clarify, the query came from the fact that the original text contained lowercase letters, whereas everywhere else in the text, it was "UBE2J1 and UBE2J2". The updated version has corrected this inconsistency in capitalisation.

Reviewer #4

(Remarks to the Author)

Response to Reviewer Comments

We thank the reviewers for the time and effort dedicated to the thorough and constructive evaluation of our manuscript. The detailed comments and suggestions have greatly contributed to improving the clarity, strength, and scope of our study. In the revised version, as detailed in the responses to individual comments and questions, we have addressed all points raised and performed several additional experiments to directly respond to the main criticisms. These include:

- Additional technical controls to exclude the possibility that membrane-responsive activity is merely the result of uneven protein reconstitution or clustering/aggregation (Reviewer 1),
- a more balanced discussion that emphasizes how our findings provide mechanistic explanations for previous observations in cellular systems, but avoids overinterpretation (Reviewer 1 and 3),
- time-resolved quantification and modelling of kinetic data, including new quenched-flow measurements (Reviewer 2),
- a more detailed analysis of the interaction of the UBC domain with the membrane, supported by molecular dynamic simulations (Reviewer 3).
- new experiments demonstrating that lipid responsiveness of the E2 directly translates into differences in substrate ubiquitination efficiency (Reviewer 3).

We believe that these additions and clarifications substantially strengthen the mechanistic conclusions, provide a clearer link between our *in vitro* findings and cellular contexts, and improve the overall clarity and presentation of the manuscript.

REVIEWER COMMENTS

Reviewer #1 (Remarks to the Author):

Endoplasmic reticulum-associated protein degradation (ERAD) are pathways of the ubiquitin-proteasome system responsible for identifying, ubiquitylating, and degrading misfolded or surplus proteins in the ER lumen and membrane. This process is essential for maintaining protein homeostasis within the ER and plays a role in regulating membrane properties by modulating the degradation of different lipid metabolic enzymes. ERAD's functionality relies on a coordinated cascade of enzymes, including E1 (ubiquitin-activating), E2 (ubiquitin-conjugating), and E3 (ubiquitin-ligase) enzymes. Recent findings suggest that ERAD's activity is influenced by the lipid composition of ER membranes; however, the mechanisms of how ERAD components sense and respond to such changes are not fully understood.

In this work, Vrentzou et al. examined the influence of membrane composition on the ubiquitylation cascade in ERAD. The team focused on E2 and E3 enzymes, particularly UBE2J2 and RNF145, reconstituting them in membranes with different lipid compositions. They reported that the E2 enzyme UBE2J2 can act as a sensor for lipid packing; its activity is impeded in loosely packed membranes, whereas tightly packed lipid environments facilitate its interaction with the E1 enzyme. Additionally, they reported that the E3 ligase RNF145 appeared to cholesterol levels, which impacts its oligomerization and enzymatic activity. Through these biochemical studies, the authors attempt to provide new insights into how lipid signals are integrated by the ERAD machinery to maintain membrane homeostasis, implicating UBE2J2 as a central regulator for lipid-responsive ERAD activity.

However, I think this manuscript falls quite short of this apparent goal and over-claims throughout. The purifications and reconstitutions of different enzymes is quite elegant, but in many cases, the experiments are without the appropriate controls to ensure that the apparent difference is actually

related to activity of the enzymes. The activity changes reported for UBE2J2 could all be explained by simple clustering, and in fact this is what my interpretation would be and their data would tend to agree. Regardless, in all cases the effects on enzymatic activity are quite small, with very dramatic lipid changes that are unlikely to ever occur, in situ.

Major concerns

Throughout the entire manuscript, there are no attempts to test any of the authors' conclusions within cells. For this reviewer, this calls the premise and conclusions into question. This fact, coupled with the unphysiological lipid compositions, make the story incomplete.

We understand the reviewer's concern regarding the lack of cellular validation. Our primary aim in this study was to analyze the direct, mechanistic effects of membrane lipid composition on components of the ERAD ubiquitination machinery, using purified components in a controlled membrane environment. This approach was chosen to eliminate the complex indirect and compensatory effects present in cells, and to dissect the fundamental biochemical principles underlying lipid-sensitive regulation.

Nonetheless, we recognize the importance of connecting these in vitro results to cellular relevance. We have therefore revised the manuscript to

- (i) temper our claims and avoid overextending inferences regarding physiological significance (see revised Abstract and Discussion, lines 486–494), and
- (ii) emphasize that our findings provide a mechanistic basis for interpreting relevant cellular observations from previous studies. For example, prior work has shown that modulation of ER lipid saturation in cells (by palmitic acid or unsaturated fatty acids) causes corresponding changes in substrate (SQLE) levels (Volkmar et al., 2022, PMID: 35993436; Stevenson et al., 2014, PMID: 24840124). Our reconstitution data provide a possible mechanistic explanation for these effects, linking membrane packing to UBE2J2 activation.

With respect to lipid compositions, we have clarified in the revised text (lines 155–159, Table 1, Figure 2e–f) that our model membranes span the range of physiologically reported saturated fatty acid content in ER membranes. UBE2J2 activity is sensitive to even modest changes in this range. In addition, we clarify in the discussion, that the full complexity of the ER membrane is not represented in our experiments (lines following L486)

We agree that further in-cell validation is desirable and will be a focus of future work. Our present findings establish direct biophysical mechanisms that can be tested in living cells.

The lipid compositions used appear to be extreme conditions that do not appear in normal cellular situations. The small, inconsistently-sized functional changes of UBE2J2 loading caused by these extreme lipid changes make the authors' interpretations difficult to agree with.

All of these effects are quite small. Technical differences could really for all of these effects. This would likely be very difficult, but I can't think of a better way to be sure than the authors should develop mutants that are non-responsive to either SFAs (for E2) or cholesterol (for E3) to ensure that these are meaningful differences.

We clarify the rationale for the lipid mixes (lines following L127): the ER is known to have a relatively high proportion of unsaturated acyl chains; published lipidomic analyses of ER membranes report saturated fatty acid (SFA) contents ranging from 25% to 40% under normal conditions (see Ref. 5, 6). Our experimental membrane compositions were designed to bracket this physiological range, with additional data extending both above and below it to provide mechanistic insight. Notably, in Figure

2e–f, we demonstrate that UBE2J2 activity responds to relatively modest and possibly physiologically relevant shifts in SFA content (from 60% down to 35%), with consistent decreases in activity observed as SFA content is lowered.

We agree that extremely high or low lipid saturation is unlikely to occur in cells, and we have revised the language throughout the text to avoid overstating the physiological significance of such extremes. Instead, our intention is to illustrate the mechanistic sensitivity of UBE2J2 to lipid packing and to demonstrate that this sensitivity is already detectable within the reported physiological range of ER membrane saturation.

With respect to the magnitude of changes in UBE2J2 loading: Our revised manuscript includes new quenched-flow experiments (Fig. 2c, d) that provide higher temporal resolution of the UBE2J2 ubiquitin loading reaction. These data reveal that loading in saturated membranes (60% SFA) occurs within 100 milliseconds to 1 second for a large fraction of UBE2J2 molecules, whereas in unsaturated membranes (10% SFA), rapid loading is almost entirely absent. This represents a striking and robust difference in loading kinetics between these membrane environments—unlike the “small, inconsistently-sized” changes the reviewer is concerned about. Our kinetic modeling (Supplementary Discussion, Fig. S10) further demonstrates how moderate shifts in SFA content impact active UBE2J2 proportions.

Furthermore, in response to the reviewer’s suggestion, we have now generated and tested a UBE2J2 mutant (F137E) predicted, based on our molecular dynamics and structure analysis, to disrupt SFA responsiveness. As shown in Figures 4i,j and Supplementary Fig. 5c,e, the F137E mutant is largely insensitive to changes in membrane saturation: it remains active regardless of SFA content and fails to adopt the auto-inhibited conformation seen in wild-type UBE2J2 under low SFA/loosely packed membrane conditions. This result strongly supports the notion that the lipid-induced changes we observe in wild-type UBE2J2 are specific, robust, and mechanistically meaningful. For E3 ligases, we agree with the reviewer that similar non-responsive mutants would provide further mechanistic resolution. However, the specific lipid-sensing elements of RNF145 and related E3s are not yet fully defined, and we have not identified E3 mutants with analogous loss of cholesterol responsiveness. Accordingly, our conclusions regarding the E3 response to lipids are necessarily less detailed and are stated more carefully, with appropriate restraint in the text.

The authors should show determine and show the orientation of the E2 in key experiments to rule out orientation as a complicating factor.

As described in the revised Methods and Results (Supplementary Fig. 1h,i; lines 117–119), the majority of UBE2J2 is inserted with the correct orientation in ER-like membranes, as confirmed by protease protection and antibody accessibility assays. These assays were performed for key experimental conditions (Supplementary Fig. 2b), confirming that the portion of UBE2J2 accessible for ubiquitin loading is very similar in different membranes.

How do the authors know that they have evenly distributed proteins between their liposomes? This would greatly affect their interpretations so it should be addressed experimentally (specifically in Figure 3D).

Similarly, the results in all cases with UBE2J2 simply point towards increased association, which could be driven by the uneven distribution of lipids within, or between, the liposomes. The crosslinking experiments would suggest this to be true.

We agree that an uneven distribution of UBE2J2 among liposomes could influence ensemble FRET measurements in Fig. 3D. In our original submission, we already took steps to minimize heterogeneity

in protein reconstitution by using controlled detergent-mediated insertion and size-extrusion protocols (Methods, “Protein reconstitution in liposomes”), ensuring homogenous vesicle size (DLS data, Suppl. Fig. 2l–m). Following reconstitution, we routinely assessed protein incorporation by flotation assay, confirming that the actual protein content per lipid was consistent between preparations and across lipid conditions.

To specifically address the concern that differences in UBE2J2 activity might be explained by uneven protein distribution (rather than true sensing of membrane packing), we performed control experiments using n-octyl- β -D-glucoside (OG) at sub-solubilizing concentrations, which tightens lipid packing (Supplementary Fig. 4h) without altering global protein distribution. Upon addition of OG to proteoliposomes with loose packing (high unsaturated lipid content), we observed

- (i) activation of UBE2J2 (restoring ubiquitin loading, Fig. 4c,d),
- (ii) a change in the limited proteolysis pattern of UBE2J2 (Fig. 4e), mirroring effects observed in tightly packed membranes.

These experiments demonstrate that the observed effects on UBE2J2 activity and conformation are due to membrane physical properties—not artifacts of uneven protein partitioning.

Concerns about uneven lipid distribution are valid. As also explained below, in response to Reviewer 2, we cannot rule out such uneven contribution for lipid mixes containing DPPC with a total SFA content of 60%. However, based on careful studies of the miscibility of various PC/PE mixtures (Silvius JR, 1986, PMID: 3707951 DOI: [10.1016/0005-2736\(86\)90350-0](https://doi.org/10.1016/0005-2736(86)90350-0); Ahn and Yun, 1999, PMID: 10486148 DOI: [10.1006/abbi.1999.1376](https://doi.org/10.1006/abbi.1999.1376)) we consider uneven lipid distribution as highly unlikely for POPC/POPE and other POPL/DOPL mixtures.

If the authors’ interpretation of Figure 2D is correct, the lipid packing would be increasing as increase OG, based on the reference cited. They should first use laurdan to test whether this is true, then, they should use their FRET assay from Figure 3D to test their interpretation.

We measured C-Laurdan fluorescence in proteoliposomes upon addition of increasing concentrations of OG (below its critical micellar concentration) to test whether OG indeed increases membrane packing under our experimental conditions. As shown in Supplementary Fig. 4h, addition of OG led to a clear blue shift in C-Laurdan emission, corresponding to an increase in lipid packing. This confirms that sub-solubilizing OG directly increases membrane packing in our liposomes.

The title does not fit, this is exaggerating the results described in this work. There is no description of How the enzymes do this task, only an indication that the activity changes. Similarly, the abstract is misrepresents which is actually observed (“master regulator”). Broadly, I think some restraint should be used to describe the work accurately.

We agree that our initial title and abstract language may have overstated the mechanistic specificity and physiological role described in this study. In the revised manuscript, we have modified both the title and abstract to more accurately reflect our results and their scope, and to avoid claims that are beyond what is mechanistically shown. We have revised the title to focus on the mechanistic finding on UBE2J2. We have removed the phrase “master regulator” and reworded statements about UBE2J2’s function so that they describe the observed effect—namely, its membrane-sensitivity and potential to integrate lipid signals as part of the ERAD cascade—without overstating broader regulatory roles. We have also reviewed the main text to ensure conclusions are appropriately stated with restraint.

Why was UBE2J1 ignored? In many cases, UBE2J1 appears to be responsive (Extended Data 2D) similarly to UBE2J2. Broadly, whether experimental results are described as significant, or not, depends on whether the results are what the authors want to claim.

Our choice to focus primarily on UBE2J2 in the present work was driven by both mechanistic and technical considerations: As shown in Figure 1d,e and Supplementary Fig. 2h–j (formerly Extended Data 2D), UBE2J1 does exhibit a modest reduction in ubiquitin loading under highly unsaturated membrane conditions. However, unlike UBE2J2, UBE2J1 loading remains rapid and efficient across all tested membrane compositions, with the correctly oriented fraction nearly fully loaded within 30 seconds (lines 127–129). The difference between different membranes may in part result from differences in membrane orientation (Supplementary Fig. 2j), and not necessarily from direct conformational inactivation akin to UBE2J2. In contrast, UBE2J2 shows a dramatic and unique sensitivity to membrane saturation and packing, with both a clear switch-like conformational response and a broad range of functional consequences (including E1 interaction, conformational states, and downstream E3 activity). We have explicitly reported the UBE2J1 results (Figure 1d,e; Supplementary Fig. 2h–j), described them in the manuscript, and clarified their interpretation: “While we did not investigate UBE2J1 kinetics or lipid sensitivity further, these results indicate that the pronounced lipid-saturation effect is specific to UBE2J2 and not a general property shared with UBE2J1” (lines 166–168). We acknowledge the reviewer’s point that statistical significance does not necessarily reflect functional relevance for every modest change, and we have avoided overstating the significance of minor effects in our text.

In sum, we focused on UBE2J2 due to its pronounced and reproducible membrane sensitivity, openly presented UBE2J1 data with appropriate caution, and have clarified the rationale for our focus in the revised manuscript.

Minor concerns

Could small variations in the activity of the enzyme be contributing to some of these differences? I think this is reasonable because there are pretty large changes of the same conditions in different figures.

We find this comment difficult to address in its current general form, as no specific comparisons between conditions or figures are indicated. We would be glad to provide a detailed explanation or perform additional analyses once the relevant datasets are specified.

Immunoprecipitations from a reconstituted system are a bit unconventional. Some additional specificity controls are probably required.

We have added Supplementary Figure 8c, which tests for unspecific binding of proteins to the bead material and for specificity of the immobilized nanobodies. With respect to the specificity of immobilizing liposomes, we have characterized this approach in previous publications: Schmidt et al., 2020, PMID: 32588820; Swarnkar et al., 2024, PMID: 39533056.

Broadly, the authors are not particularly clear with the composition of the lipid bilayer. All of these results are interpreted as differences in saturation, but oleic fatty acids and palmitoyl fatty acids also differ in chain length. The lipid compositions used fail to account for potential differences in having chains distributed between lipids, for example 50% DOPC 50% DPPC is likely to behave differently compared to 100% POPC.

Detailed lipid species and proportions are now clearly listed for each experimental condition (revised Table 1). We agree that differences in chain length and chain distribution (e.g., DOPC/DPPC vs. POPC) can affect membrane properties independent of saturation. To distinguish these effects, we included an additional control experiment in which liposomes containing DOPC/POPC/DOPE (60/20/20, 10% SFA content, and 90% of acyl chains having 18 carbon atoms) were directly compared with liposomes that contained palmitoleic acid (Po, 16:1) instead of oleic acid (DPoPC/POPC/DPoPE ; 60/20/20), that

have the same degree of saturation, but 90% of acyl chains have 16 carbon atoms. UBE2J2 reconstituted in these liposomes was equally inactive (Supplementary Figure 2d,e). Thus, we interpret the primary driver of the observed effects to be the extent of saturation, rather than chain length.

With respect to the second point of comparing liposomes with the same aggregate chain composition but different distributions (50% DOPC 50% DPPC vs. 100% POPC), there are technical limits to testing this, as mixtures with high fractions of DPPC need to be treated at rather high temperatures, which was incompatible with the protein reconstitution procedure.

Why are all proteins so large on SEC? For example, UBE2J2 is 37kDa by SDS-PAGE, broad peak 400+ kDa by SEC

Purified UBE2J2 is solubilized with detergent (in our case, DDM) during purification and SEC runs. The proteins migrate through SEC columns together with their associated detergent micelles, which significantly increases their effective hydrodynamic radius and thus causes a much larger apparent molecular mass than the polypeptide mass alone would predict.

Additionally, the broadness of the SEC peak reflects the heterogeneity in detergent-protein micelle sizes and, to some extent, the presence of different protein conformers or minor oligomerization. We have provided evidence from limited proteolysis and FRET experiments that UBE2J2 can exist as both monomers and reversible dimers depending on membrane environment and concentration.

Other suggestions

Increased labeling of Figure 3A would help readers interpreting the results.

We have adapted the figure legends and the description of the main figure in the text.

The intro focuses quite a bit on lipid properties and sensing within yeast at the transcription factor level, which is okay. But the authors have missed more recent related work on lipid influencing ERAD function, for example PMID 36638172.

We have modified the introduction accordingly (lines 52 – 65, with that specific reference in line 60)

Reviewer #2 (Remarks to the Author):

Vrentzou et al. report an important finding: E2 ligase UBE2J2 is the primary component of the ERAD response and is triggered by lipid packing density. This is a beautiful manuscript, featuring highlight-level biochemistry and crystal-clear conclusions. It also can be improved significantly on the data analysis side with relatively limited extra work (and a few additional experiments). I agree with the conclusion that the switch-like change in UBE2J2 is very likely a response to general membrane properties and not to individual lipids (as underlined by the assays in the presence of PE and cholesterol). But I also think that the data show a different regulatory mechanism in response to PE and Chol levels, which could be discussed a bit more in depth.

I am strongly in favour of publishing the paper in Nat. Commun. and recommend that the manuscript should be accepted after a few (but important) revisions.

Major point:

The authors obtain beautiful kinetics for E2 ligase loading in reconstitution assays that are far from easy. The level of biochemical ability to do these types of experiments with the time resolution that is presented here is rare indeed and should be applauded. However: The authors then proceed to simply do a linear interpolation of the data points, which is a severe under-utilization of the obtained rather fantastic data (please excuse the formulation, I want to get my point across). I strongly recommend

that the authors fit a few simple kinetic models to their data (given that the plots are made with Origin anyway, this just means copy-pasting the integrated rate law and setting the boundary conditions for the fit), which should allow them (a) to underpin the conclusions of their impressive manuscript with quantitative kinetic analysis and (b) potentially uncover new mechanistic details.

We have followed this advice and fitted a simple kinetic model to the data. However, to be able to do so, we collected more time-resolved data for the early phase of the loading reaction using a quenched-flow apparatus. These measurements revealed that in saturated membranes about half of the loadable fraction reacts within 100 ms to 1 s, whereas the rest of the protein reacts on the minute time scale. In the membrane with low SFA content, the fast-reacting pool is virtually absent. We were able to fit this data (new Fig. 2C) to the kinetic model depicted in the new Fig. 2D. The model assumes that an inactive and an active conformation of UBE2J2 are in equilibrium. This equilibrium is affected by the membrane composition. Global fitting of the data in Origin with the function derived in the Methods section (L715 – 725) showed that only k_1 (the rate for the conversion of an active to an inactive conformation) needs to be separately fit, whereas k_2 (the reverse reaction) and the pseudo-first-order loading reaction k_3 are so similar in the different membrane compositions that the fit quality was not further improved by adjusting them separately. Intuitively, such a model can be interpreted as meaning that the inactivating membrane-binding reaction is determined by the number of available binding sites.

Given that this is at saturating levels of ATP/Ub/E1 ligase, we're looking at a simple reaction which has a straightforward rate law – given the concentrations applied here it should simply be a bimolecular reaction, likely with first-order kinetics due to the high excess of reactants. By eye-balling the data, I'd say that that the corresponding rate law in its simplest form can be fitted only to one dataset: UBE2J2 after detergent treatment of the liposomes in Fig. 1.

We have not attempted to fit data collected in detergent solution, but we agree with this notion. As stated above, we treated the loading reaction as a pseudo-first-order reaction because of excess E1.

Interestingly, UBE2J1 looks to be loaded with similar kinetics (the difference is thus unlikely a question of activity), but either a large fraction of the protein remains inaccessible or loading is an equilibrium reaction/there is a significant hydrolysis rate. This can be tested by depleting ATP from the reaction mixtures after reaching the steady state and following the loaded fraction over time with the corresponding kinetic analysis. I recommend doing this experiment for both proteins.

A combination of different factors was responsible for the overall low efficiency of ubiquitin loading of UBE2J1. First, we have improved the purification procedure, so that in detergent solution now close to 80% of the protein is loaded. Second, a larger fraction of UBE2J1 than of UBE2J2 is adopting the wrong-site-out orientation upon membrane reconstitution (based on the new data of Supplementary Fig. 1n and 2j, for UBE2J1 around 50%, for UBE2J2 around 25%). Furthermore, this fraction is affected by the membrane composition in the case of UBE2J1, but unaffected for UBE2J2 (Supplementary Fig. 2b). In summary, less efficient loading of UBE2J1 is mostly due to wrong-site out orientation. We do not observe dependency on phospholipid saturation, but of course we cannot rule out that other membrane components not tested in this work might regulate UBE2J1. However, this is not a question we are concerned with in this manuscript and needs to await future studies.

We have conducted the discharge experiments for UBE2J2 (Supplementary Fig. 2g). In agreement with our previous work on Ubc6 and UBE2J2, discharge occurs on the minute time scale ($k_{\text{discharge}} \approx 0.3 \text{ min}^{-1}$, Swarnkar et al. 2024, PMID: 39533056), which means that – in our loading assays - discharge can be expected to affect UBE2J2 loading state only marginally. In the presence of an E3 ligase, a discharge rate several orders of magnitude higher can be expected and in fact has been measured for UBE2J2 in the presence of MARCHF6 (Swarnkar et al. 2024, PMID: 39533056). As we explain in the discussion and

in a supplementary discussion, continuous loading and discharge cycles are probably essential for the membrane sensitivity of UBE2J2 to have an impact on steady-state levels of loaded UBE2J2.

In all liposome datasets, the loaded fraction levels of around 0.5 which makes sense if the orientation of protein incorporation is roughly 1:1. However, I think (as do the authors judging from their formulation of “immediately available fraction for loading”) that the obtained kinetics for conditions of higher content of saturated PL cannot be explained by a simple bimolecular reaction model any more (Fig 2). The data look like two independent loading reactions (a very fast one and a slow one) overlaid as a linear combination – a clear indication that one cannot assume a well-mixed situation. Such a situation makes it difficult to draw any firm conclusion without fitting quantitative kinetic models, further underlying the need for this type of analysis.

Intriguingly, the detergent data in Fig 1C do not share this characteristic – they very much look like a single exponential could be fitted reasonably well. Thus, it is unlikely a feature of the protein itself. My guess: This is due to heterogeneity in the liposome lipid composition. A mixed initial PL composition will lead to a liposome population with a certain distribution of individual PL compositions, a fraction of which will be suitable to trigger the switch to UBE2J2 hyperactivity. Since it is unclear how addition of DPPC to a DOPE/DOPC system affects this distribution w/o having the data in hand I would probably not stress the 50% // 60% saturated side chain point overly much – there may well be nonlinear effects in play. The best model to fit here would be to assume two UBE2J2 pools of distinct activity that do not have any exchange with each other on the timescale of the experiment.

We thank the reviewer for these thoughtful comments and suggestions. We agree that data in Figure 2a, b cannot be explained by a simple bimolecular reaction. However, we don't think that the most straightforward explanation for the biphasic shape for liposomes with 50% or 60% SFAs is to treat these two phases as independent from each other, fitted by the linear combination of two exponential functions. The molecular interpretation of this would be that both states can engage with $E1 \sim Ub$, but with different rates. We think from a structural perspective it makes more sense to assume that (at least) two states of UBE2J2 exist in equilibrium with each other, but that only one of these states can engage with $E1 \sim Ub$ (that is the model shown in Fig. 2d). However, the data shown in Fig. 2c which includes the new quenched-flow data, ultimately does not allow us to differentiate between these two models.

Furthermore, we agree that for mixtures including DPPC we cannot rule out the existence of more than one liposome population with different distributions of individual lipid components and that these different liposome populations are the explanation for the biphasic behavior. However, such a scenario is highly unlikely to occur in the 50% SFA liposomes in Fig. 2a, b, which contain only POPC and POPE, and unlikely for the 40% and 35% SFA liposomes in Fig. 2e, f, which contain POPC/DOPC/DOPE mixtures. Yet, UBE2J2 reconstituted into such liposomes also exhibits biphasic Ub-loading behavior. (See also our response to a similar comment by Reviewer 1 and references given in that response on PC/PE miscibility).

The following experiments are optional, but I do believe that they would further underline & illustrate the effects of lipid composition on enzyme activity:

The distribution of lipid packing in liposomes can be experimentally tested using e.g. the single particle profiler approach that Erdinc Sezgin recently published (<https://www.nature.com/articles/s41587-023-01825-5>), which allows to measure the GP of many individual liposomes. An experiment done with proteins in GUVs at relatively low magnification (20x) with fluorescent Ub would probably lead to some GUVs lighting up immediately, while others become fluorescent only over time.

The suggested experimental setup would certainly be useful in distinguishing between the two scenarios mentioned above. However, as we think that biphasic behavior in POPC/POPE liposomes already excludes a scenario in which biphasic behavior is explained by the existence of different liposome populations, we did not further follow up on this advice. However, for future in-depth analysis of other lipid mixes this will certainly be a useful methodology.

Minor points:

- Intro: The lower SL content of the ER should also be mentioned in the characterization of ER membrane properties

Such a statement has been added (line 41)

- To me the UBE2J1 data look like there is a small but significant hyperactive fraction present under all conditions – it may well be the same conformational switch, but triggered by a different membrane property / something else.

As mentioned above, we have improved the purification protocol for UBE2J1, as judged by its generally higher activity in detergent solution.

- Line 155, should this read “PE increases packing”? The effects measured by GP and in the enzymatic assay point in different directions, if I read the data correctly. PE containing liposomes have a higher GP (higher packing) and also faster loading kinetics. Importantly, this effect is distinct from the effect of lipid saturation – it seems that PE inclusion accelerates the kinetics of the slow loading process. I am a bit confused. Please clarify

The reviewer is absolutely right, PE increases packing, as previously shown (Ballweg et al., 2020, PMID: 32029718; Dawaliby et al., 2016, PMID: 26663081) and also in Fig. 2g. In agreement with our general model that UBE2J2 senses packing, inclusion of 20 mol% PE accelerates the slow loading process, as stated by the reviewer.

- Cholesterol seems to have a quirky effect – it accelerates the slow loading reaction, but also lowers the initial jump.

We have not analyzed the effect beyond what was shown in our original submission. Inclusion of 10 mol% cholesterol modulates the behavior of UBE2J2 only slightly, and makes the difference observed for the two extreme lipid compositions somewhat smaller, as described by the reviewer.

- Please plot quantification of the data in Fig 3F/G in a time resolved fashion as in Fig. 2. Looking at the gel I do not actually think that the effect disappears. It would be rather surprising if it did. I am not at all sure that the conclusion in this section holds up, but I am sure that fitting the two models discussed above will give a yes/no answer

In the revised manuscript, we have replaced the original Fig. 3F/G with a new figure that addresses the reviewer’s core question — and the related point from Reviewer 3 — more directly.

The new data distinguish between a specific effect of increasing local UBE2J2 concentration and a potential artefact from non-specific membrane crowding. We now show that:

- Increasing UBE2J2 concentration in membranes rich in unsaturated fatty acids alleviates the activity inhibition observed under these conditions.
- In contrast, adding an equivalent amount of an unrelated membrane protein (superfolder GFP anchored via the UBE2J2 transmembrane helix) does not increase UBE2J2 activity.

- Similarly, inclusion of an enzymatically inactive UBE2J2 mutant (C94A) also counteracts inhibition.

At elevated local UBE2J2 concentration, both cysteine crosslinking and limited proteolysis patterns (Supplementary Fig. 3c and 3g) converge towards those seen in tightly packed membranes, indicating that UBE2J2 conformation becomes less dependent on membrane composition when dimerization/oligomerization are favored by higher local concentrations.

We agree with the reviewer that the functional link between dimerization and UBE2J2 reactivity needs to be considered carefully. Our data suggest that dimerization is not a strict requirement for ubiquitin loading: in Fig. 4k, sub-solubilizing concentrations of OG restore E1 interaction in 10% SFA membranes without a simultaneous increase in crosslinking, indicating that the correlation between dimerization/oligomerization and E1 reactivity is not absolute.

- It would have been good to include the 6s timepoint in the experiment with the UBE2J2/Syb2 chimera in Fig 4a.

- Same for Fig 4e. Data are convincing, but would have been even more so with additional time resolution.

We agree with the reviewer that additional early time points can provide useful insight into the initial phase of the reaction. For the revised manuscript, the early phase kinetics of wild-type UBE2J2 are now characterised in detail by quenched-flow experiments (Fig. 2c,d), which directly address this time regime with high temporal resolution.

Repeating the full set of experiments for the UBE2J2/Syb2 chimera (Fig. 4a, now Supplementary Figure 4c) or for the experiment in Fig. 4d,e (now Fig. 4c,d) with similar temporal resolution would have required setting up equivalent quenched-flow measurements for each protein construct. Given the technical demands and time investment such an extension entails, we have not re-run these experiments.

We emphasise that the overall conclusions from the chimera (and the additionally provided J2-J1 chimera) and OG-activation experiments are based on robust differences at later time points, which align well with the high-resolution kinetics obtained for the wild-type enzyme. We believe this provides a strong and internally consistent basis for our interpretations.

- I think that the detergent addition experiment does not conclusively address the packing defects point. That would be more convincingly done by showing the recruitment of a fluorescent biosensor for packing defects (e.g. a fluorescently labelled ALPS motive)

We appreciate the suggestion and agree that ALPS-motif-based probes are well-established reporters for membrane packing defects. We believe, however, that there may be a misunderstanding regarding the aim of our OG-addition experiment.

Our goal was not to measure packing defects directly, but rather to test whether conditions known to reduce hydrophobic interactions could restore UBE2J2 activity. Sub-solubilizing concentrations of OG are reported to intercalate into defect sites and tighten lipid packing (Heerklotz, 2008). In our experiments, OG addition to low-SFA membranes both altered UBE2J2's tryptic accessibility toward the "tightly packed" pattern and restored E1 interaction (Fig. 4c–e,k, and Supplementary Fig. 4i), consistent with the idea that association with packing defects leads to inactivation.

While a fluorescent ALPS motif would indicate the presence of defects in the membrane, it would not by itself demonstrate that such defects control UBE2J2 activity, nor directly link their modulation by OG to the functional and conformational changes we observe. For these reasons, we consider the

current combination of OG perturbation with biochemical and proteolysis readouts to be a more direct test of the proposed mechanism.

- With regard to the dimerization model of activation, the authors should check whether the faster loading kinetics cannot simply be explained by the higher enzyme concentration and amend Figure 7 and the discussion if necessary.

We note that in the experiments shown in the revised Fig. 3f,g, the total concentration of (wild type) UBE2J2 in the reaction mixture was kept constant across conditions. Thus, in the case of wild type UBE2J2, less proteoliposomes of higher P/L ratios were added, but protein-free liposomes were added in addition to match total lipid concentrations. In the case of sfGFP-TMS_{J2} and the catalytically dead UBE2J2, the same amount of proteoliposomes were added. Thus, the faster loading kinetics observed at high P/L in unsaturated membranes cannot be explained by a simple bulk enzyme concentration effect.

André Nadler

Reviewer #3 (Remarks to the Author):

Overview

In this manuscript, the authors investigate the membrane sensing properties within components of the mammalian ERAD mechanism using a reconstituted ubiquitylation system comprised of purified E3 and E2 enzymes embedded in proteoliposomes of varying composition. In doing so, the authors reveal several different levels at which membrane composition influences the ubiquitylation cascade. Most notably, they provide evidence that indicates the E2 enzyme UBE2J2 contains a region/domain within its cytoplasmically exposed portion, that effectively senses lipid packing – where loosely packed lipids containing low fraction of saturated fatty acids (10%) appeared to impede loading of activated ubiquitin onto the E2 UBE2J2 by the E1. When investigating the impact of phospholipid saturation, the authors found that an environment of tight packing (50-60% SFA) in contrast, permitted engagement and ubiquitin transfer by the E1. The loss of activity arising from slight adjustments to the SFA composition (40%>30%) would seem to indicate a degree of sensitivity with UBE2J2. Adding cholesterol slightly increased activity when SFA content was lower (35%) but did not add anything when it was higher (60%). The responsiveness to different compositions are used as support a hypothesis that UBE2J2 is a sensor for lipid packing. Limited proteolysis of UBE2J2 under different lipid compositions and release of an N-terminal fragment indicates a conformational change taking place. Evidence from TMD swapping is provided that sensing of lipid packing occurs through the cytosolic domain and is attributable to interaction/s between hydrophobic domain/s of UBE2J2 and membranes with “packing defects”. As UBE2J2 works together with ERAD E3s that regulate sterol/lipids (e.g. MARCHF6, RNF139, RNF15), the authors suggest that this built in regulation helps to control membrane composition and that UBE2J2 sits at the centre as a “master regulator”. Evidence is provided that each E3’s auto-ubiquitylation was lipid dependent and attributable to UBE2J2, just as interaction with the E1 appears to be. Collectively, the authors make a case for UBE2J2 as key component in the regulation of ER lipid composition through intrinsic features sensitive to local changes that impact the ubiquitylation cascade.

Overall impression

Mechanistic insight into membrane embedded ubiquitylation machinery participating in ERAD through reconstitution is a welcome addition to the plethora of genetic and cell-based data that has defined our understanding for some time. It should be of general interest both in its strategy and in its findings. This manuscript does a very good job of executing this technical study and in doing so, reaching some

interesting conclusions about the potential interdependency of ubiquitylation machinery and ER membrane composition. It represents an advancement in the field as it highlights a potentially novel point of feedback on membrane composition. Overall, the manuscript is well written, the technical experiments performed to a high standard, the data of good quality. In most cases the appropriate statistical analyses have been included where necessary and are described. There were however several instances where the replicates were not sufficient (e.g. n=2). For the most part, the Methods and Materials describe the experiments that have been performed.

The main issues this reviewer has with the manuscript arise in two areas – 1) the certainty for which the term “master regulator” is used for UBE2J2 when the supporting evidence only partially addresses such a broad role and 2) the absence of any details regarding the “hydrophobic” interaction governing UBE2J2 and loosely/defectively packed lipids. For the former, while the impact changing lipid composition/packing has on UBE2J2 is clear, what is not clear is whether such a reductionist approach is sufficient to permit the use of the term “master regulator”, when for all intents and purposes, it’s nothing more than a conformational change that permits (or restricts) accessibility. Inclusion of some cell-based assays when lacking UBE2J2 and/or combined with treatments biasing ER lipid composition would go some way to earning the term “master regulator”. Regarding the latter, there is some concern within this reviewer that the domain/region of UBE2J2 that is purportedly acting to sense the membrane is not described in detail. The authors demonstrate the TMD (~aa 227-247) is not responsible. If not the UBC domain of UBE2J2 comprising the first ~160aa at the N-terminus, this leaves approximately ~60aa where a “sensor” might reside. Constructing chimeras with UBE2J1 could offer a straightforward approach to identifying the residues/domain involved. Alternatively, providing a Alpha Fold model highlighting hydrophobic patches within UBE2J2 (that maybe are not there in UBE2J1) could also help to support the authors conclusions.

On the first major criticism, we agree that the evidence for UBE2J2 as a “master regulator” primarily derives from reconstituted in vitro systems and that the cell-based dimension is not directly demonstrated in this manuscript. Our intention was to convey that UBE2J2 potentially acts as a key regulatory node within the ERAD cascade, governing the responsiveness of multiple E3 ligases to changes in membrane composition. In light of the Reviewer’s comment, we have revised the manuscript to make this reasoning more explicit. In the revised Discussion, we more clearly delineate the limits of our experimental system and explicitly state that future work will be essential in confirming in a cellular context whether UBE2J2 alone is important for regulating ERAD activity in response to dynamic changes in lipid composition.

Regarding the second major criticism, we have added a substantial body of work to address the lack of mechanistic detail. First, as suggested by the reviewer, we have constructed chimeric versions of UBE2J1 and UBE2J2 (Fig. 4 and Supplementary Figure 4). These chimeras identify the UBC domain as necessary for UBE2J2’s sensitivity to membrane saturation. To gain more detailed insight into potential sites in the UBC domain interacting with the membrane, we conducted molecular dynamic simulations, which identified a cluster of hydrophobic residues on the UBC domain surface as potential contributors to membrane association (Fig. 4f-h and Supplementary Fig. 5). We performed targeted mutagenesis of F137 (generating F137E and F137Y), and our functional and conformational assays demonstrate that the F137E substitution abolishes lipid sensitivity, consistent with this aromatic residue being essential for membrane engagement when packing defects are present.

Nevertheless, the manuscript is an advancement of understanding and a new vantage point for the how membrane composition can impact ERAD. There are several comments/suggestions that this reviewer has regarding the data presented. They are listed below.

Queries & Comments

1. For Figure 1B, Could the authors provide an explanation for the multiple bands observed with UBE2J2DL680 in detergent? There is some mention of loading and auto-ubiquitylation in the legend of Extended Data Fig 1D but this is not discussed anywhere else in the text and it is not clear why E1 transfer of Ub would produce such a pattern.

The observed pattern reflects attachment of multiple ubiquitin moieties to UBE2J2. However, the data in Fig. 1B alone do not reveal whether these represent multiple ubiquitin modifications on different cysteine residues (UBE2J2 contains 4 Cys residues), or the formation of short ubiquitin chains on a single cysteine. To address this, we generated a UBE2J2 mutant containing only the catalytic cysteine. As shown in Supplementary Fig. 1f, g, this single-cysteine variant shows almost no attachment of multiple ubiquitins, indicating that the multiple bands observed for wild-type UBE2J2 arise from monoubiquitination at several cysteine residues, rather than ubiquitin chain formation on one site. In contrast, when the catalytic cysteine is mutated to alanine, virtually no ubiquitin attachment is seen, indicating that initial ubiquitin loading onto the catalytic cysteine is a prerequisite for subsequent transfer to other residues in UBE2J2. This behavior is consistent with what has been described for the *S. cerevisiae* homolog Ubc6, where multi-monoubiquitination also occurs, though in that case serine, threonine, and lysine residues serve as acceptors (Swarnkar et al. 2024, PMID: 39533056; Schmidt et al., 2020, PMID: 32588820).

Finally, we note that this multi-band pattern is observed almost exclusively in detergent solution. In the membrane-reconstituted condition, auto-ubiquitination is largely suppressed (or only appears after very long incubations) and the main band corresponds to the E2~Ub thioester. We attribute the increased auto-modification in detergent to greater conformational flexibility and unrestricted access of E1/Ub to all potential sites.

2. For Figure 1D, there is no corresponding reducing (R) lane for UBE2J1DL680 as there is for 1B. Was this an oversight?

This has been added (Fig. 1d)

3. In the text, the authors often refer to ER-like membranes. Could the authors speak to the heterogeneity of lipid composition within the ER membrane, it's uniformity (or lack there of) within cells and speculate how this might impact localization of ubiquitylation machinery in it. More specifically, are the E2 and E3 enzymes partitioning in selected parts of the ER as a way to control/limit or engage their activity? Could the authors speculate whether it is the "active" or "inactive" state that UBE2J2 would preferentially/routinely reside in?

On the heterogeneity of the ER membrane: Although we refer to "ER-like membranes" in our study to describe synthetic bilayers formulated to approximate the average lipid composition of the ER, it is documented that the ER membrane exhibits lipid heterogeneity *in vivo*, even though this has not been investigated as deeply as for the plasma membrane. Recent studies using lipidomics and imaging have revealed that the ER comprises microdomains with variable content of saturated and unsaturated phospholipids, cholesterol, and sphingolipids (Ardail et al., 2003, PMID: 12578562; Shen et al., 2017, PMID: 29196526; Reinhard et al., 2024, PMID: 38491296). This heterogeneity may reflect functional subdomains such as ER exit sites (Weigel et al., 2021, PMID: 33852913; Runz et al. 2006, PMID: 16794576) or membrane contact sites (Fujimoto et al. 2012, PMID: 22185692) and could fluctuate dynamically according to cell type or metabolic state.

On the consequences for ubiquitination machinery localization: Given that our *in vitro* data show UBE2J2 activity is strongly dependent on local membrane packing (i.e., lipid saturation), it is plausible that, in the cellular context, E2 and E3 enzymes may preferentially partition into, or be retained in, ER subdomains with appropriate lipid composition. This partitioning could be governed by intrinsic affinity

for certain membrane properties (e.g., tight packing, higher levels of saturated lipids or cholesterol) or by protein-protein interactions with other ERAD components or adaptors. Such a mechanism would allow the cell to spatially and temporally restrict ubiquitination activity—potentially preventing unwanted substrate modification under basal conditions or activating ERAD in response to local changes in lipid composition. Based on our model and available lipidomics data, the “average” ER membrane has a relatively high proportion of unsaturated lipids, which in our reconstituted system stabilizes the autoinhibited, membrane-associated conformation of UBE2J2. Therefore, we speculate that, under normal conditions, the majority of UBE2J2 molecules may reside predominantly in the “inactive” state, limiting non-specific ubiquitination and providing a pool that can be activated by local increases in lipid packing (e.g., during metabolic stress, in specific ER microdomains, or in response to signaling cues).

In summary, both the uniformity (average) and heterogeneity of the ER lipid landscape likely play a critical role in determining where and when ERAD machinery is active. We have added a necessarily brief discussion of these points to the revised manuscript, which also point out that our reconstitution approach lacks this full-complexity of the native ER (L486-494).

4. In Figure 3 and on pg 7, line 187-188 – Is there any *in vivo* or physiological evidence that UBE2J2 forms a dimer/oligomer? Moreover, while increasing the protein concentration of UBE2J2 appears to favour its activation, could this not just be attributable to non-specific crowding? Could the authors for example titrate in a catalytically dead UBE2J1 to drive home the point that UBE2J2 multimerization may important?

First, regarding *in vivo* or physiological evidence of UBE2J2 dimerization/oligomerization: To our knowledge, there is currently no direct evidence for stable dimer or oligomer formation of UBE2J2 under physiological conditions in cells. Our findings that UBE2J2 dimerization is favored by high membrane concentration and enhanced by tightly packed membranes are based on *in vitro* reconstituted systems (Fig. 3c–e). Therefore, while our data suggest that dynamic self-association could contribute to regulation, we cannot confirm that stable dimers or oligomers of UBE2J2 are present *in vivo*.

Second, regarding the effect of high membrane protein concentration: We agree that the observed rescue of UBE2J2 activity at high local concentration could be attributed not only to specific multimerization but also to general effects of membrane crowding. To address specificity, we compared full-length UBE2J2 to a TMD-GFP construct. Only the full-length protein was able to overcome membrane-induced inactivation at high concentration (Fig. 3f,g), indicating that simple membrane anchoring and crowding via the TMD are not sufficient. Furthermore, we tested whether catalytic activity is required for this rescue by using a catalytically inactive UBE2J2 mutant at high membrane concentration. We observed that this mutant also overcame membrane-induced inactivation, indicating that the effect is independent of catalytic turnover and further supporting the idea that proximity-induced conformational effects (rather than enzymatic activity *per se*) are responsible.

We did not perform the suggested experiment with UBE2J1, because we think the comparison shown in Fig. 3f,g sufficiently addresses the question, but it remains an interesting avenue for future work. Our domain-swap data suggest that unique features in UBE2J2’s cytosolic region are required for activity rescue, but formally testing the specificity of multimerization with non-cognate E2s would provide additional clarity.

5. What domains of UBE2J2 are driving/influencing dimerization? Is it the same region responsible for sensing lipid packing? UBE2J1 chimeras might find some use here.

Our data show that dimerization/oligomerization of UBE2J2 is promoted in membranes with high saturated fatty acid content and by high local protein concentration, as demonstrated by FRET and crosslinking (Fig. 3c–e, Fig S3C). In addition, when we increased the protein-to-lipid ratio (P/L) in the membrane, only full-length UBE2J2—and not a construct consisting of the UBE2J2 TMD fused to GFP—could overcome membrane-induced inactivation (Fig. 3f,g). This indicates that membrane clustering through the TMD alone – if occurring at all – is insufficient and that the cytosolic region, including the UBC domain, is essential for restoring activity under conditions of loose lipid packing.

While this underscores the crucial role of the cytosolic domain, we have not directly measured dimerization propensity for UBE2J2/UBE2J1 domain-swap mutants in our study. The activity-based assays with these chimeras (Fig. 4a,b) demonstrate that swapping the TMD or the disordered linker does not abolish lipid sensitivity, suggesting that these regions are not essential for sensing. However, direct experiments comparing dimerization of these chimeras or mutants (e.g., via FRET or crosslinking) have not been performed. Therefore, we cannot formally exclude possible contributions from the TMD or linker to dimerization.

6. The TMD of UBE2J2 was swapped with that of Syt and sensitivity to lipid saturation was still observed. While demonstrating that sensitivity was not attributable to the sequence itself, UBE2J2 and Syt TMDs both have predicted hydrophobicities on the higher side (of the Kyte-Doolittle scale) for TA proteins. If a TMD from an ER-resident TA protein with lower hydrophobicity were to be swapped (like CYB5A or SQS), would the expectation still be that what senses packing was independent of membrane anchoring?

This is a valid point. In our study, we have only swapped the TMD of UBE2J2 with those of Synaptobrevin (Syb)2 and, as new data, UBE2J1 (Fig. S4c and 4a,b, respectively), and observed that membrane sensitivity is preserved in both cases. To assess how TMD hydrophobicity might affect lipid sensing, we compared the predicted free apparent free energy of membrane insertion (ΔG_{app}) using the ΔG prediction server v1.0 (<https://dgpred.cbr.su.se/index.php?p=home> Ref: Hessa et al., 2007, PMID: 18075582; while all compared proteins are not or unlikely to be inserted by the Sec translocon, we think that comparison of the computed ΔG_{app} values still gives a good indication of the relative hydrophobicities). This analysis indicates that the TMD of UBE2J2 is substantially less hydrophobic than those of Syb2 and UBE2J1 (-0.931 vs. -4.478 and -3.186, respectively) and more closely resembles those of SQS and CYB5A (-0.241 and -0.263, respectively).

Based on these results, we do not expect the use of a TMD from SQS or CYB5A—both with lower hydrophobicity—to abolish membrane-sensitive ubiquitin-conjugating activity, since UBE2J2's native TMD is already relatively low in hydrophobicity. However, while our chimeric constructs show that neither the specific sequence nor the degree of hydrophobicity of the TMD is strictly required for sensitivity, we cannot fully exclude a contribution from the general physicochemical properties shared by these TMDs or their influence on UBE2J2's conformation and function. Therefore, as noted in our revised manuscript (L249), we have qualified our conclusion as follows: *We conclude that the UBC domain is primarily responsible for sensing lipid packing, though general physicochemical properties shared by the disordered linkers of UBE2J1 and UBE2J2 or between the tested transmembrane segments may also contribute.*

7. For Figure 5, is auto-ubiquitylation of the E3 an valid readout for a physiological process? Would it be technically feasible for a “substrate” to be introduced as well and ubiquitylation of it be used to assess the ubiquitylation cascade?

We think that E3 auto-ubiquitination is a valid and informative readout for measuring the functional cooperation between specific E2/E3 pairs and for determining whether increased E2 activity translates into higher E3 activity. However, we also concur with the reviewer that reconstituting a more

physiological ubiquitination reaction provides stronger and more relevant conclusions. In response, we have established a substrate ubiquitination assay using squalene monooxygenase (SQLE), a well-characterized endogenous substrate of MARCHF6 and UBE2J2. The results, shown in Fig. 5e-g, confirm that membrane-saturation dependent activity of UBE2J2 dictates downstream ubiquitination efficiency. We would also like to point out that lower levels of SQLE have been observed upon exposing cells to palmitic acid (Volkmar et al., 2022, PMID: 35993436), and that treatment of cell with different unsaturated fatty acids stabilized SQLE (Stevenson et al., 2014, PMID: 24840124). These observations support our model that membrane sensing and membrane dependent activity of UBE2J2 influence degradative outcome.

8. In Figure 5D an line 297, the authors state that the soluble UBE2J2 without a TMD was insensitive to membrane saturation. Yet the authors infer that the cytosolic portion of UBE2J2 that is hydrophobic is what senses lipid packing/saturation. How can this be resolved? Is membrane tethering a prerequisite if not a determinant of UBE2J2 activation?

Our data indicate that membrane tethering via a transmembrane segment is indeed a prerequisite for UBE2J2 activation in response to membrane saturation. While the cytosolic portion of UBE2J2 contains hydrophobic regions that may interact with membranes, these interactions alone are insufficient unless the protein is membrane-anchored. This suggests that the transmembrane domain is necessary to localize the UBC domain in close proximity to the membrane, thereby allowing its hydrophobic regions to effectively sense changes in lipid packing/saturation. We believe this is due to the relatively low affinity of the cytosolic region for membrane lipids, which only becomes functionally relevant when a high local concentration is achieved via membrane integration.

9. Regarding the reducing conditions of the samples, in the figure legends it is stated as 2% (v/v) 2-mercaptoethanol (R) although in the M&M it is written 6%. Which one is correct?

Thank you for pointing out this mistake. The sample buffer stock is 3x and not 4x as stated before. The reducing version of the 3x stock contains 6% β -mercaptoethanol, resulting in 2% in the actual sample mix. This has been fixed throughout the text. The formulation of the 3x stock is given in L668.

Typos and Text inconsistencies

- L38: membrane protein biogenesis 3-6, lipid biosynthesis

Fixed the komma (now L40)

- L186: in which Ube2J2 was active (Extended Data Figure 3A).

Fixed (now L211)

- L376: thus loading of Ube2J2 with ubiquitin.

Fixed but rephrased (now L455)

- L431: *S. cerevisiae*

Fixed (now L524)

- L437: need space after 66

Fixed (now L532)

- L438: 4000 x g

Fixed (now L532), also on other occasions

- L442: lysate was cleared by

Fixed (now L536)

- L443: 4000 x g, which temperature?

Fixed (now L537)

- L450: (w/v) detergent ◊ which one?

This is specified for each individual protein, e.g. DDM for UBE2J2, but DM for AUP1

- L470: missing .

Fixed (now L563)

- L471: remove were

Fixed (now L565)

- L480: through a microfluidizer (?)

Detailed information about the instrument used has been added (L574).

- L488: (w/v) detergent ◊ which one?

As stated above, this is specified for each individual protein.

- L545: ultracentrifugation at 55000 rpm ◊ all the others are given in x g

Fixed (now L647), also fixed the wrong rotor name

- L565: 5 mM MgAc2) for 30 minutes

- L566: 5 mM MgAc2)

Throughout the manuscript this has been replaced with “magnesium acetate”.

- L578 & L586: with or without 6% (v/v) β-mercaptoethanol ◊ the Fig. legends, it is written 2%

Thank you for pointing out this mistake. The sample buffer stock is actually 3x and, when reducing, contained 6% β-mercaptoethanol, resulting in 2% in the actual sample mix. This has been fixed throughout the text. The formulation of the 3x stock is given in L668.

- L667 and L671: β-MCD

Fixed (L816 and L818)

- L675: 2500xg

Fixed throughout

- L677: NaCl,

Fixed (L828)

- L686 (table 5): pET39 His14-SUMO-UBE2J1S184D-TMUBE2J2-LPETGG

Fixed. Transmembrane segments in this table are now more clearly defined by residue numbers (now Supplementary Table 5)

- Figure 1: Panel B and D: font size different for ER-like and detergent.

Fixed

Panel C and E: Loaded vs modified?

Now consistently called modified throughout.

L2: Ube2J1 and Ube2J2 both possess an N-terminal

We don't understand this criticism. If referring to the singular form of "an N-terminal UBC domain" we think this is correct, as it refers to a singular feature present in each protein. However, we are not native English speakers.

- Figure 2: L4: deviation.(C) As in (A), (missing space)

Fixed

L9: ..described in 14.(F) Ubiquitin loading (missing space)

Fixed

- Figure 4: Panel C and D: [OG] in mM vs [OG] (mM)

Fixed to "[OG] in mM"

L3: acids (SFAs).(B) (missing space)

- Ext Data Fig 2: L6: scanning.(C) Ubiquitin...(missing space)

L14: described in 14.(H) Dynamic... (missing space)

We hope that all missing spaces have been added .

Reviewer #4 (Remarks to the Author):

"I co-reviewed this manuscript with one of the reviewers who provided the listed reports. This is part of the Nature Communications initiative to facilitate training in peer review and to provide appropriate recognition for Early Career Researchers who co-review manuscripts."

Thank you for your time and effort. We really appreciate it.

We thank the reviewers for their constructive feedback and insightful comments, which have helped us improve the clarity, accuracy, and contextualization of our work. We greatly appreciate their time and expertise in evaluating our manuscript.

REVIEWERS' COMMENTS

Reviewer #1 (Remarks to the Author):

In revision, the authors have significantly strengthened their manuscript and have addressed my major concerns- where possible with important data.

I appreciate the restraint from overclaiming in the rewriting of the manuscript, this has made the story more accurate. I think the authors should try to include some of the context from the questions raised by the reviewers in the discussion that were unable to be addressed. I don't think it takes away from the importance of the work, it just provides some more clarity for readers. For example around some uncertainty with protein and lipid distributions between liposomes.

We have added the following comment in the Discussion to accommodate the reviewers concern more explicitly "In addition, the exact distribution of proteins and lipids among individual liposomes is not directly controlled. Although we adjusted protein-to-lipid ratios and used defined lipid mixtures, stochastic variation between vesicles may generate local environments that differ from the average composition. Such variation could influence quantitative aspects of the reactions, such as local protein crowding, but is unlikely to affect the qualitative dependencies on lipid packing and cholesterol observed here." (Line 488)

But, I have no further substantial concerns and I believe this work should be published.

Reviewer #2 (Remarks to the Author):

The authors sufficiently addressed my points and I support the publication of the revised manuscript
André Nadler

Reviewer #3 (Remarks to the Author):

In this revised manuscript, the authors have addressed most of this reviewers concerns. 2 minor points.

1. In line 72, the authors state that "UBE2J2 and UBE2J1 also modify serine and threonine residues thus expanding the substrate repertoire" and cite 2 papers to support this statement. While valid for UBE2J2, neither of these papers contain data demonstrating UBE2J1 has this capacity. While it might be anticipated, no formal evidence has been published. Perhaps this sentence could be reworked to reflect this subtle distinction.

We agree that the evidence for non-canonical ubiquitination by UBE2J1 is weaker than for UBE2J2. Besides the homology, the notion is largely based on work by Paul Lehner (new Reference 46). This paper showed that a lysine-free HLA variant is ubiquitinated via the HRD1 pathway, that the modification is sensitive to NaOH treatment (indicating serine or threonine modification), and requires the presence of UBE2J1. Unlike for UBE2J2, there is no mass-spectrometric evidence or a more detailed enzymological analysis of UBE2J1. We have added this reference and express the weaker experimental basis for noncanonical ubiquitination as follows:

“For example, UBE2G2 mediates formation of Lys48-linked ubiquitin chains and modifies substrate lysines⁴⁵, whereas UBE2J2 – and probably UBE2J1 – additionally modify serine and threonine residues, thus expanding the substrate repertoire^{24,39,46}.”

2. Our initial review queried a line in Figure 1, Line 2.... "Ube2J1 and Ube2J2 both possess an N-terminal, catalytic..." which the authors were confused about and stated in their rebuttal. To clarify, the query came from the fact that the original text contained lowercase letters, whereas everywhere else in the text, it was "UBE2J1 and UBE2J2". The updated version has corrected this inconsistency in capitalisation.

Addressed already. Thanks for the clarification.

Reviewer #4 (Remarks to the Author):
